# User Response in Ad Auctions: An MDP Formulation of Long-term Revenue Optimization

## ABSTRACT

We propose a new Markov Decision Process (MDP) model for ad auctions to capture the user response to the quality of ads, with the objective of maximizing the long-term discounted revenue. By incorporating user response, our model takes into consideration all three parties involved in the auction (advertiser, auctioneer, and user). The state of the user is modeled as a user-specific click-through rate (CTR) with the CTR changing in the next round according to the set of ads shown to the user in the current round. We characterize the optimal mechanism for this MDP as a Myerson's auction with a notion of modified virtual value, which relies on the value distribution of the advertiser, the current user state, and the future impact of showing the ad to the user. Leveraging this characterization, we design a sample-efficient and computationally-efficient algorithm which outputs an approximately optimal policy that requires only sample access to the true MDP and the value distributions of the bidders. Finally, we propose a simple mechanism built upon second price auctions with personalized reserve prices and show it can achieve a constant-factor approximation to the optimal long term discounted revenue.

## KEYWORDS

Ad auctions, Reinforcement learning, MDP

### ACM Reference Format:

Anonymous Author(s). 2018. User Response in Ad Auctions: An MDP Formulation of Long-term Revenue Optimization. In *Proceedings of Make sure to enter the correct conference title from your rights confirmation emai (Conference acronym 'XX)*. ACM, New York, NY, USA, 16 pages. https://doi.org/XXXXXXX.XXXXXXX

## 1 INTRODUCTION

Auctions have proven to be highly robust mechanisms for price discovery and for optimizing revenue or social welfare. Recent high impact applications of auctions include auctions for internet advertising [16, 42] and wireless spectrum allocations [27]. The classic design and analysis of auctions naturally focuses on the outcomes for two parties: the auctioneer and the bidders, i.e., the seller and the buyers. The seminal works of Vickrey [43], Clarke [13], and Groves [21] give auction mechanisms which guarantee an outcome with the optimal *social welfare*. Another seminal work of

Myerson [33] gives the design of optimal auctions, i.e., those which maximize *revenue* for the auctioneer.

However, in many applications, there are additional parties involved in the transaction. We are specifically interested in the domain of internet advertising auctions. Besides the auction platform and the advertisers, an important party involved in the auction is the *user* who is viewing and interacting with the ads on the search results page. The business objective of the auction is to provide relevant ads to the user. For example, in sponsored search, the user is issuing a search query with some intent of purchasing a good or a service, and the goal of the advertising system is to connect the user to sellers who can provide the desired goods or services.

The key metric for capturing this business objective is the *long-term* social welfare or revenue across millions of repeated auctions. Most prior works on ad auctions (see exceptions in Section 1.2), focus on single-shot auctions and do not incorporate user response to ad quality, thus are unable to capture any sort of long-term effects. As two examples, a user may see completely irrelevant ads, or see seemingly relevant and useful ads which turn out to be malware. Both these examples result in very poor experiences to the user and the user may no longer interact with any ads shown in future. This effect has been nicely captured by Hohnhold et al. [24] in a paper which established the empirical importance of showing higher quality ads. They established that user satisfaction is driven by the quality of ads viewed or clicked in the past and described an experimental design which measures long-term effects on the users' propensity to click on ads: they show, based on real-world and large-scale experiments, how low-quality ads can lead to *ads blindness*, i.e., the user will stop interacting with ads in the future, even if the future ads are relevant and of good quality. Similarly, high quality ads can lead to *ads sightedness*. We discuss other related work in Section 1.2.

In this paper, we capture this important empirical observation via a theoretical model which is amenable to an auction design analysis. Specifically, we propose a model based on a Markov Decision Process (MDP) to capture the user's response to the quality of the ad. With this model in mind, we design an auction that uses these signals to obtain an optimal (or approximately optimal) auction in terms of the long-term revenue.

**Our Model.** We model the setting (details in Section 2) as a repeated interaction between a user, who is modelled using an MDP, and an ads system, who is the decision maker of the MDP. In each auction, ad candidate $i$ comes with a bid $b_i$. The user state is modeled via their click-through-rate (CTR), which we think of as the user's propensity to click.[1] We assume that the auctioneer knows, or can estimate, the impact on the user's CTR when a set $W$ of ads is shown to the user. In Section 5, we further assume that in each auction, ad candidate $i$ also comes with a signal $q_i$. While our

---

[1]The user state can be considered as a multiplier for the ad-specific CTR. Note that, in this paper, we hide the ad-specific CTR into the value of the ad.

theory in Section 5 does not require the signal to have any semantic meaning, we will think of $q_i$ as a quality signal.

## 1.1 Our Results

In this model, we first provide (Section 3) a characterization of the long-term (discounted) revenue-optimal auction which balances both the (short-term) revenue considerations per round and the positive or negative longer-term effects of showing a good or bad ad. A well-known result in the reinforcement learning literature is that the auction that optimizes the long-term discounted revenue must satisfy a recursive equation known as the Bellman Equation and that the optimal auction can be found using an algorithm such as value iteration [37]. Naively, this would require that we optimize a particular function over the (infinite-sized) space of all possible auctions. Thus, it is not a priori clear whether or not this optimization problem is even tractable.

Interestingly, we show that the long-term revenue-optimal auction takes a recognizable form. A seminal result due to Myerson [33] showed that, when bidders' valuations are drawn from some regular distribution, the revenue optimal auction maximizes *virtual welfare*, which is a function of both the bid and the value distribution of each bidder. In our model, we define the notion of a *modified virtual welfare* which consists of the original virtual welfare plus a correction term that takes into account the long-term impact of showing a particular set of ads. This correction term is calculated based on the MDP formulation and depends on the current user state and the set of ads shown. We show that the long-term revenue-optimal auction is the one which maximizes this modified virtual welfare in each round, and prices the ads accordingly. In other words, the optimal auction is a Myerson auction with modified virtual welfare. One immediate benefit of such a characterization is that, in the single-slot setting, the optimization problem in the Bellman Equation now becomes not only tractable but indeed has a closed-form solution.

We next consider the question of whether we can *learn* an approximately optimal mechanism when we do not know the MDP or bidders' value distributions. More specifically, we consider the generative model where one is allowed sample access to the MDP for any given state-outcome pair. We show that the problem can be essentially decoupled so that we can separately learn the MDP transitions and the value distributions. While our techniques are inspired by existing literature on learning optimal policies for MDPs and learning approximately optimal mechanisms, we note that there are some technical differences that we need to handle. First, the majority of prior work on learning optimal policies assume that the state transition depends on the action taken where the action set is either finite or exhibits certain linear structure. Here, the set of actions correspond to the set of all possible mechanisms which is an infinite set with complex structure. However, the set of possible outcomes, where an outcome corresponds to the set of shown ads, is finite, and we make use of this observation to show that it is possible to adapt existing results. Second, most results for learning approximately optimal mechanisms rely on the structure of the revenue-optimal auction. However, recall from the previous paragraph that the mechanism that optimizes the long-term revenue maximizes the *modified* virtual welfare in each round and

thus, is not a per-round optimal mechanism and depends on the MDP learning question. Indeed, these two learning problems are highly inter-dependent. For this reason, existing results are not directly applicable but we show how to adapt existing results to our setting.

Finally, we follow the spirit of the "simple versus optimal" literature [23] in seeking a mechanism whose structure is similar to a second-price auction that can approximate the long-term revenue under user response. This has two key benefits. First, the pricing and allocation is more transparent to the advertisers. Second, from the auctioneer's perspective, this reduces the design space to help make the auction tuning more tractable in practice. One particular implementation of this, which is similar to the one that will be discussed in this paper, is to first filter out all bidders that do not meet their personalized reserve. We then allocate to the highest bidder in the auction and they are charged the higher of their personalized reserve and the second second-highest bid. Hartline and Roughgarden [23] show that this simple auction with the appropriate reserves (in particular, the Myerson reserve) achieves a 2-approximation of the optimal revenue in the single-shot setting. In our setting, we explore what can be achieved using auctions that fall into this family of second-price auction with personalized reserves.

Our third result (Section 5) shows that there is indeed a version of a second-price auction with personalized reserves which provides a constant factor approximation to the long-term revenue-optimal auction from Section 3. A technical challenge in designing such an auction is that the auction may cause the state transition of the user to behave very differently than the optimal auction. In particular, if we use such an auction, we need to utilize the personalized reserve prices to control the user state transitions and use this as a proxy to trade-off the current round revenue with the long-term impact. Ideally, we would like for two things to be true. First, we want that the personalized reserves that we introduce induces user state transitions that are very similar, or even identical, to the user state transitions of the optimal mechanism. Second, we would like to guarantee that, at each step of the auction, the personalized reserves are chosen in a way that the revenue is a constant factor approximation to the revenue obtained using the long-term revenue-optimal auction. Indeed, it turns out that it is possible to achieve both desiderata simultaneously.

To summarize, our work generalizes classical auction theory to the setting of long-term revenue optimization with user response. We bring together the reinforcement learning theory and auction theory by using an MDP to capture the user response. Additionally, we characterize the optimal mechanism for this setting by combining the Bellman Equation and Myerson's auction. Building upon this characterization, we design algorithms that learn approximately optimal mechanisms from samples. Furthermore, we extend ideas from the simple versus optimal mechanism design literature to our setting.

## 1.2 Related Work

A number of other models have been studied in the literature that incorporate some form of user response. Athey and Ellison [4] and Linden et al. [29] consider a single-stage position auction where

a user has some budget and clicking on an advertisement incurs some search cost and consumes a portion of this budget. Some relevant literature, including Abrams and Schwarz [1], Bachrach et al. [7], Lahaie and Pennock [26], Schroedl et al. [38], Thompson and Leyton-Brown [41] incorporate the user's externality in determining ad *placement*. While these papers focus on some form of user response for a single query, our model is focused on understanding user response *across* queries. Another point of view of this paper is that we use an MDP to capture the effect of future user response created by showing an ad in the current round, and incorporate that into the auction design.

In another set of related works, Li et al. [28], Stourm and Bax [40] use "shadow costs" or "hidden costs" to capture the effect of negative user experiences on the ad platform's future revenue. They design good auctions to maximize long-term revenue assuming these costs are given. Ashlagi et al. [3] consider a mathematical model where there may be multiple search engines competing for users. The user has a cost for user a search engine depending on a search cost and also their "distance" from a search engine. They study equilibria of search engines in this setting. In this paper, we use an MDP model to provide microfoundations for these costs (or gains) and show how to learn the optimal auction without knowing the costs (or gains) a priori.

Another stream of related work includes the problem of dynamic mechanism design which has been heavily studied in the literature (e.g. [5, 9, 12, 25, 30, 31, 35, 36]; see also the surveys [8, 10] and the references therein). Some of these works also consider an MDP setting [25, 36]. For example, Kakade et al. [25] consider a setting where the *advertisers'* values evolve according to a Markov process, although their specific setting requires separability assumptions that do not capture our setting involving the user. A key differentiator between these sequences of works on dynamic mechanisms is that they assume the *advertisers* are long-lived and they evolve over time, say via an MDP. Thus, the advertisers are assumed to satisfy a long-term incentive compatibility constraint which themselves resemble Bellman Equations (for example, see Section 2 of [10]). On the other hand, we assume that advertisers are static[2] and our model assumes that it is the *user* that evolves over time. As a result, we only need to guarantee that the auction is incentive compatible for each individual round which allows us to have a particularly clean characterization of the optimal mechanism.

## 2 MODEL AND PRELIMINARIES

In this section, we describe the model we consider in this paper. We assume that we are in a repeated auction setting with $n$ advertisers (bidders) competing for one of $k$ identical ad slots of a query from a single user in each round.[3] At each round $t$, the advertiser $i \in [n]$ has a value $v_i$ drawn from a (regular) distribution $\mathcal{F}_i$ with CDF $F_i$ and PDF $f_i$ whose support is bounded in $[0, 1]$. In Section 3, we assume that showing a set $W$ of ads to the user has a known affect on the user; this is discussed more in the next paragraph. In Section 5, we assume instead that each ad $i$ comes with a signal

$q_i \in \{-1, 1\}$. Although $q_i$ need not have any semantic meaning, we refer to it as a *quality* signal where $q_i = 1$ means the ad is good and $q_i = -1$ means the ad is bad. We note that the value distributions and quality signals may change over time but since most of our analysis focus on a single point in time, we omit a subscript on the time. For simplicity, we assume the advertisers are myopic. In other words, we assume that each advertiser aims to optimize their utility only in the current auction and does not try to optimize their utility across all auctions. We believe this is a valid assumption since the advertisers do not know the identity of the user and the advertisers may be involved in many other auctions all involving different users. In fact, an advertiser may not even see the same user twice.[4] Thus, it may be difficult for an advertiser to directly benefit from any specific user response.

A novel contribution of the paper is to provide a natural, yet general, model of the effect of the ad quality on the user's responsiveness to ads in the future. We model the user effects of showing good or poor ads using a Markov Decision Process (MDP). Suppose that the user, at round $t$ has a *user-specific* click-through rate (or propensity to click), $\text{ctr}_t$,[5] which is used to measure the probability that the user is willing to click an ad that is shown to her. Without loss of generality,[6] we assume $\text{ctr}_t$ is independent of the ad shown to the user. If $W \subseteq [n]$ is the set of ads that are shown then we assume that that the next round click-through-rate $\text{ctr}_{t+1}$ is drawn from some distribution denoted by $P_W(\cdot|\text{ctr}_t)$. In the case of single-slot auctions, we will also use $P_i$ to denote $P_{\{i\}}$ for $i \in [n]$ and $P_0$ to denote $P_\emptyset$. For our results in Section 5, which focuses on single-slot auctions, we do make a further simplifying assumption that assumes the ctr in the next round $\text{ctr}_{t+1}$ depends only on $\text{ctr}_t$ and $q_i$ and not on the identity of the ad. We also assume that quality is binary. Intuitively, this means an ad is either good or bad.

With this user model in mind, we can then describe the repeated auction setting as an MDP $\mathcal{M} = (\mathcal{S}, \mathcal{A}, P(\cdot|a, \text{ctr}), R(a, \text{ctr}))$ where

- The set of states $\mathcal{S}$ is the interval $[0, 1]$. We represent a state by ctr which denotes the user-level click-through rate.
- The set of actions $\mathcal{A}$ is the set of all mechanisms that the auctioneer may use. In this paper, we restrict the auctioneer to using incentive compatible mechanisms (see Subsection 2.1 for some background in auction theory).
- $P(\cdot|a, \text{ctr})$ is the transition probability which determines the ctr of the user at the next state. We assume it is given as follows. Let $\mathcal{D}^a$ be the distribution of winners for the auction $a$ when the value distributions are $F_1, \ldots, F_n$. Then $P(\cdot|a, \text{ctr}) = \mathbb{E}_{W \sim \mathcal{D}^a} [P_W(\cdot|\text{ctr})]$. In particular, note that the transition is independent of the auction given which ads are shown.[7]
- $R(a, \text{ctr})$ is the (expected) reward function. Given a truthful mechanism $a$ with allocation $x^a(v) \in [0, 1]^n$ where $\sum_{i=1}^n x_i^a(v) \leq k$ and (expected) payment function $p^a(v) \in \mathbb{R}_{\geq 0}^n$, we have that $R(a, \text{ctr}) = \text{ctr} \cdot \mathbb{E}\left[\sum_{i=1}^n p_i^a(v)\right]$. Note

---

[2]Our model doesn't require the advertisers are fixed at each round as long as we know the set of the advertisers.

[3]Our analysis never makes use of the fact that the number of advertisers are the same so we can allow the number of advertisers to be different in every round. For simplicity, we will assume that the number of advertisers in each round is the same.

[4]For example, if a user submits two different search queries, it is quite likely that the set of advertisers for the two queries are disjoint.

[5]For simplicity, we assume that all the slots are associated with the same click-through rate. Our results also hold when the click-through rate is different for each slot.

[6]Our model allows the ad-specific click-through rate, but we hide this factor in the value $v_i$ of each ad $i$.

[7]Note that $P$ without a subscript will correspond to the transition of the chain given an action (i.e., auction) and $P_X$ denotes the transition when the ad set $X$ is shown.

that $(x^a, p^a)$ must be chosen in an incentive compatible manner (see Subsection 2.1).

Finally, we assume an infinite-horizon MDP setting and we let $\gamma < 1$ be the discount factor. For a policy $\pi \colon \mathcal{S} \to \Delta_{\mathcal{A}}$, the long-term discounted revenue starting at state $\mathrm{ctr}_0$ is given by

$$V^\pi(\mathrm{ctr}_0) = \mathbb{E}\left[\sum_{t=0}^{\infty} \gamma^t \mathop{\mathbb{E}}_{a_t \sim \pi(\mathrm{ctr}_t)} [R(a_t, \mathrm{ctr}_t)]\right], \tag{2.1}$$

where $\mathrm{ctr}_{t+1} \sim P(\cdot|a_t, \mathrm{ctr}_t)$. Note that a distribution over IC mechanisms is itself an IC mechanism, so $\Delta_{\mathcal{A}} = \mathcal{A}$. Thus, without loss of generality, given a state $\mathrm{ctr}_t$, we assume that the policy $\pi$ outputs a single (possibly, randomized) mechanism and so the policy is actually deterministic. Finally, we note that we can extend all our results to the finite-time horizon setting with the caveat that our policies may depend on the number of rounds $t$ so far.

## 2.1 Standard Auction Preliminaries

In this work, we focus on incentive compatible (IC) mechanisms[8], i.e., reporting true value is the optimal bidding strategy for each bidder no matter how the other bidders bid. For each IC auction $a \in \mathcal{A}$, we have an allocation rule $x^a \colon \mathbb{R}_{\geq 0}^n \to [0, 1]^n$ and payment function $p^a \colon \mathbb{R}_{\geq 0}^n \to \mathbb{R}_{\geq 0}^n$, where $\sum_{i=1}^n x_i^a(v) \leq k$. Here, $x_i^a(v)$ and $p_i^a(v)$ represent the allocation probability and (expected) payment of bidder $i$ respectively when the reports (bids) are $v$ (we may write $v$ as the bids since the auctions are assumed to be IC). Given Myerson's Lemma [33], for any IC mechanism $a \in \mathcal{A}$, allocation $x^a(\cdot)$ is monotone with the input bid and the payment rule satisfies the payment identity $p^a(b) = b \cdot x^a(b) - \int_0^b x^a(v)\,\mathrm{d}v, \forall b \in \mathbb{R}_{\geq 0}$. Therefore, we can rewrite the expected reward function $R(a, \mathrm{ctr})$ as $R(a, \mathrm{ctr}) = \mathrm{ctr} \cdot \mathbb{E}_v \left[\sum_{i=1}^n x_i^a(v)\phi_i(v_i)\right]$ where $\phi_i(v) := v - \frac{1 - F_i(v)}{f_i(v)}$ is the virtual value function for each advertiser $i$. Throughout this paper, we assume value distribution $\mathcal{F}_i$ is regular so that $\phi_i(\cdot)$ is non-decreasing for each bidder $i$. Moreover, since we focus on IC mechanisms, we slightly abuse the notation to use allocation rule $x$ to represent the mechanism and the payment rule can be induced by $x$ following Myerson's Lemma [33]. Throughout the paper, we interchange advertiser and bidder to refer to the same entity.

## 3 OPTIMAL MECHANISM

In this section, we characterize the optimal policy in our MDP setting. Recall that the policy maps each state to an IC mechanism $a \in \mathcal{A}$. Following the standard notations in infinite-horizon MDP literature, let $V^*$ denote the value function under the optimal policy, i.e. $V^*(\mathrm{ctr}) = \sup_\pi V^\pi(\mathrm{ctr})$ for all $\mathrm{ctr} \in [0, 1]$, where $V^\pi$ is defined in Eq. (2.1). For notation, for a set $W \subseteq [n]$, we let $P_W(\cdot|\mathrm{ctr})$ denote the transition probability function if we show the ads in $W$ when the state is $\mathrm{ctr}$ and $P(\cdot|a, \mathrm{ctr})$ denotes the overall transition probability (over the randomness in value distributions) when the auction is $a$ and the state is $\mathrm{ctr}$.

**THEOREM 3.1.** *Suppose that the advertiser distributions are regular and that there are $k$ identical slots. In each state $\mathrm{ctr}$, the optimal (IC)*

---

[8]In the single-parameter setting, Bayesian incentive compatibility (BIC) is equivalent to dominant strategy incentive compatibility (DSIC). So we use "IC" or "truthful" interchangeably in this paper, depending on context.

*mechanism allocates the slots to the advertisers in the set $W \subseteq [n]$ with $0 < |W| \leq k$ that maximizes*

$$\mathrm{ctr} \cdot \sum_{i \in W} \phi_i(v_i)$$

$$+ \gamma \left(\mathop{\mathbb{E}}_{\mathrm{ctr}' \sim P_W(\cdot|\mathrm{ctr})} \left[V^*(\mathrm{ctr}')\right] - \mathop{\mathbb{E}}_{\mathrm{ctr}' \sim P_\emptyset(\cdot|\mathrm{ctr})} \left[V^*(\mathrm{ctr}')\right]\right), \tag{3.1}$$

*provided that Eq. (3.1) is positive (otherwise, the mechanism does not allocate).*

The proof of Theorem 3.1 is deferred to Appendix A. It is instructive to compare this mechanism with Myerson's optimal auction. Recall that Myerson's auction chooses the set $W$ that maximizes the virtual welfare, i.e. $\sum_{i \in W} \phi_i(v_i)$; this latter term is precisely the first term of Eq. (3.1) (modulo the ctr term which is independent of the set $W$). Theorem 3.1 asserts that to optimize the discounted long term-revenue, it suffices to incorporate an additive correction term to the virtual welfare, i.e. the second term of Eq. (3.1). Intuitively, the correction term can be thought as the long-term revenue impact of showing a set of ads compared to a baseline of not showing any ads.

**REMARK 3.2.** *Regularity is not strictly necessary for Theorem 3.1. Indeed, if the distributions are not regular then one may always use Myerson's ironing procedure [33] to the modified virtual value, so that the resulting ironed modified virtual value is monotone in the bidder's value.*

## 3.1 Single-slot auctions

In this subsection, we show that we can further simplify the form of the optimal mechanism presented in the previous section when there is only a single slot. Recall that, for $i \in [n]$, we let $P_i(\cdot|\mathrm{ctr})$ denote the transition probability function if we show ad $i$ when the the state is $\mathrm{ctr}$, and let $P_0(\cdot|\mathrm{ctr})$ denote the transition probability function if no ads are shown.

**DEFINITION 3.3 (MODIFIED VIRTUAL VALUE).** *Fix an advertiser $i \in [n]$ whose value is drawn from a distribution $F_i$. We define the modified virtual value as*

$$\widetilde{\phi}_i(v_i, \mathrm{ctr}) = \mathrm{ctr} \cdot \phi_i(v_i) +$$

$$\gamma \left(\mathop{\mathbb{E}}_{\mathrm{ctr}' \sim P_i(\cdot|\mathrm{ctr})} \left[V^*(\mathrm{ctr}')\right] - \mathop{\mathbb{E}}_{\mathrm{ctr}' \sim P_0(\cdot|\mathrm{ctr})} \left[V^*(\mathrm{ctr}')\right]\right).$$

Following Theorem 3.1, we immediately have,

**COROLLARY 3.4.** *Suppose that the advertiser distributions are regular. In each state $\mathrm{ctr}$, the optimal (IC) mechanism allocates to the advertiser that maximizes the modified virtual value $\widetilde{\phi}_i(v_i, \mathrm{ctr})$ if at least one of the modified virtual values is positive (otherwise, the mechanism does not allocate).*

## 4 LEARNING APPROXIMATELY OPTIMAL POLICIES

In the previous section, we developed the optimal mechanism when we have access to the valuation distribution of the bidders and the transition probability matrix of the MDP. In this section we present approaches to design *approximately*-optimal policies (mechanisms) which require only *sample* access to the valuation distribution of

the bidders and the transition matrix; this is known as the *generative model* in the Reinforcement Learning literature. We start with an intermediate setting where the valuation distributions are known, but the auctioneer only has sample access to the transition matrix of the MDP. In particular, we assume that for every state-outcome pair $(ctr, W) \in \mathcal{S} \times [n+1]^k$ the learner can obtain samples from $P_W(\cdot | ctr)$. In this setting, we design an efficient algorithm that computes an $\varepsilon$-optimal policy with probability $1 - \delta$ using poly$(1/(1-\gamma), 1/\varepsilon, \log(1/\delta), |\mathcal{S}|, n^k)$ samples from the MDP. Its running time is polynomial in the number of samples.

Subsequently, we show how to handle the case where both the MDP and the valuation distributions are unknown. We present an algorithm that computes an $\varepsilon$-optimal policy with probability at least $1 - \delta$ using poly$(1/\varepsilon, 1/(1-\gamma), \log(1/\delta), |\mathcal{S}|, n^k)$ samples from the MDP and poly$(1/\varepsilon, 1/(1-\gamma), \log(1/\delta), |\mathcal{S}|, n, k)$ samples from the valuation distributions. Moreover, the running time of the algorithm is poly$(1/\varepsilon, 1/(1-\gamma), \log(1/\delta), |\mathcal{S}|, n^k)$. The omitted results and proofs of this section can be found in Appendix B.

### 4.1 Learning Approximately Optimal Policies: Known Valuations, Unknown MDP

In the setting where the MDP (transition matrix) is unknown and the valuation distributions are known, our approach is conceptually simple: we use a large enough number of samples to estimate an empirical MDP and then we compute an approximately optimal policy with respect to that MDP. We can show that, with high probability, that policy will also be approximately optimal with respect to the true MDP. This approach is inspired by a line of work in the RL literature (see, e.g. [2, 19] and references therein). The main technical challenge is to handle the fact that the action space of the auctioneer is infinite. However, the key insight is that the transition of the MDP depends on a finite number of different outcomes of the auction. In this section, we will use $\widehat{P}$ to denote an empirical MDP. We let $V^\pi$ and $\widehat{V}^\pi$ be the value function of the policy $\pi$ with respect to the true MDP and empirical MDP, respectively. Further, we let $V^\star$ and $\widehat{V}^\star$ denote the optimal value function with respect to the true MDP and empirical MDP, respectively.

**Lemma 4.1 (Performance of Policies in Empirical MDPs).** *Consider a repeated $k$-slot auction among $n$ bidders. Let $\mathcal{W} \subseteq [n+1]^k$ be the set of potential outcomes of the auction. Let $\mathcal{S}$ be the state space. Let $\widehat{P}$ be the empirical MDP that is constructed using $N$ samples from each state-outcome pair. Then, for every policy $\pi$ and any $\delta > 0$, with probability at least $1 - \delta$ over the random draw of the samples it holds that*

$$||V^\pi - \widehat{V}^\pi||_\infty \leq \frac{2\gamma\varepsilon_{opt}}{1-\gamma} + 3\frac{\gamma^2}{(1-\gamma)^3} \cdot \sqrt{\frac{2\log(2|\mathcal{S}||\mathcal{W}|/\delta)}{N}}$$

$$||V^\star - \widehat{V}^\star||_\infty \leq \frac{\gamma}{(1-\gamma)^2} \cdot \sqrt{\frac{2\log(2|\mathcal{S}||\mathcal{W}|/\delta)}{N}}$$

*where $\varepsilon_{opt} = ||\widehat{V}^\star - \widehat{V}^\pi||_\infty$.*

The proofs uses ideas from [2, 19] that are generalized to our setting with an infinite action space.

The following result, whose proof is deferred to Appendix B, is an adaptation of the main result of [39] which handles infinitely dimensional action spaces and allows for $\varepsilon$-greedy policies.

**Theorem 4.2 (From Value Estimation to Policy Estimation (Adapted from [39])).** *Let $\widetilde{V} \in \mathbb{R}^\mathcal{S}$ be a value function such that $||\widetilde{V} - V^\star|| \leq \varepsilon_{opt}$. Let $\widetilde{\pi} : \mathcal{S} \times \mathcal{A}$ be an $\varepsilon'$-greedy policy with respect to $\widetilde{V}$, i.e.,*

$$R(ctr, \pi(ctr)) + \gamma \sum_{ctr'} \mathop{\mathbb{E}}_{W \sim \mathcal{D}^{\pi(ctr)}} [P_W(ctr'|ctr)] \cdot \widetilde{V}(ctr') \geq$$

$$\max_{a \in \mathcal{A}} R(ctr, a) + \gamma \sum_{ctr'} \mathop{\mathbb{E}}_{W \sim \mathcal{D}^a} [P_W(ctr'|ctr)] \cdot \widetilde{V}(ctr') - \varepsilon', \forall ctr \in \mathcal{S}.$$

*Then $||V^\star - V^{\widetilde{\pi}}||_\infty \leq \frac{2\gamma\varepsilon_{opt} + \varepsilon'}{1-\gamma}$.*

Finally, we will make use of the following forklore result. For completeness, we provide a short proof in Appendix B adapted to our setting where the transition of the MDP does not depend directly on the action that was taken.

**Theorem 4.3 (Approximate Bellman Update).** *Let $V^{(0)}(ctr) = 0, \forall ctr \in \mathcal{S}$. For every $k \geq 1$ define*

$$V^{(k)} = \max_{a \in \mathcal{A}} R(ctr, a) + \sum_{ctr' \in \mathcal{S}} \mathop{\mathbb{E}}_{W \sim \mathcal{D}^a} [P_W(ctr'|ctr)] \cdot V^{(k-1)}(ctr') + \varepsilon_{k-1},$$

*where $||\varepsilon_{k-1}||_\infty \leq \varepsilon$. Let $\pi_k$ be an $\varepsilon'$-greedy policy with respect to $V^{(k)}$, i.e., for every $ctr \in \mathcal{S}$ it holds that*

$$R(ctr, \pi_k(ctr)) + \sum_{ctr' \in \mathcal{S}} \mathop{\mathbb{E}}_{W \sim \mathcal{D}^{\pi_k(ctr)}} [P_W(ctr'|ctr)] \cdot V^{(k)}(ctr') \geq$$

$$\max_{a \in \mathcal{A}} R(ctr, a) + \sum_{ctr' \in \mathcal{S}} \mathop{\mathbb{E}}_{W \sim \mathcal{D}^a} [P_W(ctr'|ctr)] \cdot V^{(k)}(ctr') - \varepsilon'.$$

*Then,*

$$||V^{\pi_k} - V^\star||_\infty \leq \frac{2\gamma}{(1-\gamma)^2} \cdot \left(\gamma^k + \varepsilon + \frac{(1-\gamma)\varepsilon'}{2\gamma}\right).$$

Equipped with the previous results, we can show the following corollary, whose proof is postponed to Appendix B.

**Theorem 4.4.** *Let $\delta \in (0, 1), \gamma \in (0, 1), \varepsilon \in (0, 1)$. Then, there is an algorithm that given full access to the valuation distributions of the bidders and $N = O\left(\frac{\log(|\mathcal{S}| \cdot |\mathcal{W}|/\delta)}{(1-\gamma)^6 \varepsilon^2}\right)$ samples from the true transition kernel of every state-outcome pair outputs a policy $\widehat{\pi}$ such that, with probability at least $1 - \delta$, $||V^\star - V^{\widehat{\pi}}||_\infty \leq \varepsilon$. The running time of the algorithm is poly$(|\mathcal{S}|, |\mathcal{W}|, n, 1/\varepsilon, \log(1/\delta), 1/(1-\gamma))$*

### 4.2 Learning Approximately Optimal Policies for Single-Slot Auctions: Unknown Valuations, Unknown MDP

We now proceed to the more challenging setting where both the MDP and the valuation distributions of the bidders are unknown to the seller. Similarly as before, our approach is divided into two steps: we first use a sufficiently large number of samples from the generator to estimate the transition probability of the MDP, and then we compute an approximately optimal policy with respect to the empirical MDP. The main challenge is to deal with the fact that the reward function $R(ctr, a)$ is unknown. In the traditional RL literature this is not an issue since the sample complexity of estimating the transition probability accurately is larger than the sample complexity required to estimate the reward function. However, in our case there is an infinite number of actions so estimating $R(ctr, a)$

for every $(\text{ctr}, a) \in \mathcal{S} \times \mathcal{A}$ is non-trivial. Hence, we take a different approach that combines core RL algorithms, like value iteration, with results regarding the sample complexity of estimating revenue optimal single-parameter auctions (see, e.g. [11, 14, 15, 20, 22, 32]). The crucial observation is that, using Theorem 3.1 we can view every step of the value iteration algorithm as a single-parameter *modified* revenue-maximization problem, where we take into account both the current round revenue and the future revenue of the auction. This result also shows that the optimal auction in the $k$-th iteration for every state $\text{ctr} \in \mathcal{S}$ is the one that, given any valuation profile $v \in [0, 1]^n$ as input, allocates the slots to the bidders that maximize Equation (3.1), assuming that this quantity is non-negative, otherwise it does not allocate the slots.

An important step of our approach is to design approximately optimal auctions with respect to the *modified* revenue objective and it is inspired by the work of Devanur et al. [15]. Let us first provide the formal definition of this objective.

DEFINITION 4.5. *Let $\gamma$ be the discount factor of the MDP, let $V : \mathcal{S} \rightarrow [0, 1/(1 - \gamma)]$ and $\text{ctr} \in [0, 1]$. Denote $M$ as a truthful mechanism for a $k$-slot auction with allocation rule $x^M(v) \in [0, 1]^n$ and payment rule $p^M(v) \in \mathbb{R}_{\geq 0}^n$. Let $\mathcal{F} = F_1 \times \ldots \times F_n$. The modified revenue objective with respect to $\text{ctr}$ and $V$ is defined to be*

$$\text{ctr} \cdot \underset{v \sim \mathcal{F}}{\mathbb{E}} \left[ \sum_{i=1}^{n} p_i^M(v) \right] +$$

$$\gamma \sum_{\substack{W \subseteq [n] \\ |W| \leq k}} \underset{\text{ctr}' \sim P_W(\cdot | \text{ctr})}{\mathbb{E}} \left[ V(\text{ctr}') \right] \cdot \underset{v \sim F}{\mathbb{E}} \left[ \sum_{i \in W} x_i^M(v) \right].$$

The approach of [15] consists of two main steps. First, they show that by "rounding" the distribution of the bidders to multiples of $O(\varepsilon/k)$ the revenue of the optimal mechanism with respect to the rounded distribution will be close to the revenue of the optimal mechanism with respect to the true distribution. Then, using a uniform convergence result they show how to compute an approximately optimal mechanism with respect to the modified distribution. An important difference in our setting is that the future revenue term that appears in Definition 4.5 depends on the *set* of bidders that are selected by the auctioneer, and cannot be decomposed across individual bidders. The straightforward adaptation of the approach in [15] to handle the modified revenue objective induces sample complexity of the order poly($n^k$), since there are $n^k$ different sets of bidders that need to be considered. We are able to circumvent this issue by adding an extra step in this approach which consists of a discretization in the *virtual* valuation space.

An important technical tool that we use is a concentration bound that first appeared in [6] and was also used in [15, 20]. The version of the bound we use is the one from [15] and its formal statement can be found in Lemma B.1 in Appendix B. Essentially, this result shows that, for a sufficiently large number of samples, the expected revenue of any *fixed* auction with respect to the uniform distribution on the samples is close to its expected revenue with respect to the true distribution. In order to get a uniform convergence result we need to restrict the number of different auctions we consider. To that end, we first show that discretizing the valuation space will not change the modified revenue of the optimal auction too much. The

proof is an adaptation of Lemma 6.3 in [15] that allows us to handle the extra term in the modified revenue objective. For completeness, we provide a short proof in Appendix B

LEMMA 4.6 (ADAPTATION OF LEMMA 6.3 [15]). *Given any product value distribution $\mathcal{F} = F_1 \times \ldots \times F_n$ where $F_i$ is supported on $[0, 1], \forall i \in [n]$, and any $\varepsilon > 0$, let $\widehat{\mathcal{F}}$ be the distribution that is obtained by rounding the values of $\mathcal{F}$ to the closest multiple of $\varepsilon$ from below. Let $\widetilde{\text{OPT}}(\mathcal{F}), \widetilde{\text{OPT}}(\widehat{\mathcal{F}})$ be the optimal modified revenue with respect $\mathcal{F}, \widehat{\mathcal{F}}$. Then, we have $\widetilde{\text{OPT}}(\widehat{\mathcal{F}}) \geq \widetilde{\text{OPT}}(\mathcal{F}) - k \cdot \varepsilon$, where $k$ is the number of slots.*

The next step, which is the point of departure from [15], is to consider the class of *threshold mechanisms*. Intuitively, these are mechanisms that round down the virtual values of the bidders to the closest multiple of some given resolution $\beta$. This idea is inspired by the concept of $t$-level auctions that appeared in [32].

DEFINITION 4.7. *Let $\mathcal{F} = F_1 \times \ldots \times F_n$ be the valuation distribution of $n$ bidders, where $F_i$ is supported on $[0, 1], \forall i \in [n]$. Let $\mathcal{S}$ be the state space of the MDP, $\gamma$ be the discount factor, $\text{ctr} \in [0, 1]$ be the click-through rate, $k$ be the number of slots and $V : \mathcal{S} \rightarrow [0, k/(1-\gamma)]$ be a value function. We say that an auction $M$ is a threshold auction at resolution $\beta$ with respect to the modified revenue objective if:*

- *For every input $v = (v_1, \ldots, v_n)$ and every bidder $i \in [n]$ the mechanism uses a non-decreasing mapping $\sigma_i : [0, 1] \rightarrow \{-\infty\} \cup \left\{ \frac{-2k}{1-\gamma}, \frac{-2k}{1-\gamma} + \beta, \ldots, 1 \right\}$.*
- *The mechanism allocates to the set of bidders $W$ with $0 < |W| \leq k$ that maximize*

$$\text{ctr} \cdot \sum_{i \in W} \sigma_i(v_i)$$

$$+ \gamma \left( \underset{\text{ctr}' \sim P_W(\cdot | \text{ctr})}{\mathbb{E}} \left[ V(\text{ctr}') \right] - \underset{\text{ctr}' \sim P_\emptyset(\cdot | \text{ctr})}{\mathbb{E}} \left[ V(\text{ctr}') \right] \right),$$

*if this quantity is non-negative, and does not allocate to any bidder otherwise.*

*The payment rule of the auction is the one that makes it truthful, i.e., follows Myerson's payment rule.*

We now show that there is a threshold auction at some appropriate resolution whose modified revenue is close to the modified revenue of the optimal auction.

LEMMA 4.8. *Let $\mathcal{F} = F_1 \times \ldots \times F_n$ be the valuation distribution of $n$ bidders, where $F_i$ is supported on $[0, 1], \forall i \in [n]$. Let $\phi_i : [0, 1] \rightarrow (-\infty, 1]$ be the (ironed) virtual valuation function of bidder $i$. Let $\mathcal{S}$ be the state space of the MDP, $\gamma$ be the discount factor, $\text{ctr} \in [0, 1]$ be the click-through rate, $k$ be the number of slots and $V : \mathcal{S} \rightarrow [0, k/(1-\gamma)]$ be a value function. Let $\varepsilon > 0$. Then, there is a threshold auction $M$ at resolution $\varepsilon/k$ which has modified revenue $\widetilde{\text{Rev}}(M, \mathcal{F})$ that satisfies*

$$\widetilde{\text{Rev}}(M, \mathcal{F}) \geq \widetilde{\text{OPT}}(\mathcal{F}) - \varepsilon,$$

*where $\widetilde{\text{OPT}}(\mathcal{F})$ is the optimal modified revenue with respect to $\mathcal{F}$.*

We next show how we can get uniform convergence result for auctions that have this structure, when the valuation distributions are discrete. Crucially, by doing the discretization in the virtual value space we are able to avoid the dependence on $n^k$ when we take the union bound over all the possible threshold mechanisms in our uniform convergence argument.

LEMMA 4.9. *Given any product value distribution $\mathcal{F} = F_1 \times \ldots \times F_n$ where every $F_i$ has support $B \subseteq [0, 1]$, and any $\varepsilon > 0, \delta > 0$, there is an algorithm that takes as input $m$ i.i.d. samples from $\mathcal{F}$ and outputs an auction whose modified revenue is at least $\widetilde{OPT}(\mathcal{F}) - \varepsilon$, with probability at least $1 - \delta$, whenever*

$$m = \widetilde{O}\left(\frac{n \cdot |B| \cdot k^4}{\varepsilon^2 (1-\gamma)^2} \log(1/\delta)\right),$$

*where $\widetilde{OPT}(\mathcal{F})$ is the optimal modified revenue with respect to $\mathcal{F}$. Moreover, the running time of the algorithm is $\text{poly}(m, |\mathcal{S}|)$.*

Combining the results we have discussed so far, we show how to construct an approximately optimal mechanism when the valuation distributions are continuous. The formal statement of the result can be found in Theorem B.2 in Appendix B. The algorithm, which appears in Algorithm 2 in Appendix B, consists of the following main steps. First, it draws a large enough number of samples from $\mathcal{F}$. Then, it rounds down all the samples to multiples of $O(\varepsilon/k)$ and considers the empirical distribution $\widehat{F}$ on the samples. Subsequently, it computes the (ironed) virtual values $\widehat{\phi}_i$ with respect to $\widehat{\mathcal{F}}$. Given any input profile $v = (v_1, \ldots, v_n)$, it first rounds down the values of each $v_i$ to multiples of $O(\varepsilon/k)$, denoted by $\widehat{v}_i$, and estimates the virtual values $\widehat{\phi}_i(\widehat{v}_i)$. Finally, it rounds down $\widehat{\phi}_i(\widehat{v}_i)$ to multiples of $O(\varepsilon/k)$, denoted by $\widehat{w}_i$, and allocates the slots to the set of bidders $W$ that maximize the modified virtual welfare objective with respect to the values $\widehat{w}_i$, assuming that this objective is non-negative. It is worth mentioning that we can output a description of this mechanism in polynomial time in the parameters of the problem, however running it and computing its modified revenue requires time $\text{poly}(n^k)$.

We are now ready to present our approximately optimal policy estimator. The high-level idea of our approach it to estimate an approximately optimal policy with respect to the empirical MDP using the value iteration algorithm, where in every step we perform an approximately optimal update using the result from Theorem B.2. We are able to show that these approximately optimal updates suffice in order to end up with an approximately optimal policy.

THEOREM 4.10. *Let $\varepsilon, \delta > 0$ be the error bound and the confidence bound, respectively. Let $\gamma$ be the discount factor of the MDP and $n$ be the number of bidders. Then, Algorithm 1 given $M = \widetilde{O}\left(\frac{|\mathcal{S}| \cdot n \cdot k^5}{\varepsilon^3 (1-\gamma)^{12}} \log(1/\delta)\right)$ samples from the valuation distribution of the bidders $\mathcal{F} = F_1 \times \ldots \times F_n$ and $N = O\left(\frac{\log(|\mathcal{S}| \cdot |\mathcal{W}|/\delta)}{(1-\gamma)^6 \varepsilon^2}\right)$ samples from the generator of the MDP for every state-output pair $(\text{ctr}, W) \in \mathcal{S} \times \mathcal{W}$, outputs a policy $\widehat{\pi}$ which, with probability at least $1 - \delta$ satisfies*

$$||V^{\pi} - V^{\star}||_{\infty} \leq \varepsilon.$$

*Moreover, the running time of the algorithm is polynomial in the number of samples.*

The proof follows by combining results we have discussed so far and can be found in Appendix B.

# 5 CONSTANT-FACTOR APPROXIMATION SIMPLE MECHANISM

As discussed in Section 3, the revenue-optimal mechanism is a Myerson's auction with modified virtual value. However, Myerson's

---

**Algorithm 1** Approximately Optimal Policy Estimator from Samples

1: **Input**: Accuracy parameter $\varepsilon$, confidence parameter $\delta$, discount factor $\gamma$, number of bidders $n$, sample access to the generator of the MDP, sample access $\mathcal{F} = F_1 \times \ldots \times F_n$ where $F_i$ is valuation distribution for bidder $i$.

2: **Output**: A policy $\widehat{\pi} : \mathcal{S} \rightarrow \mathcal{A}$ with the guarantee that, with probability $1 - \delta$, $||V^{\pi} - V^{\star}||_{\infty} \leq \varepsilon$.

3: $K \leftarrow O\left(\frac{\log(1/((1-\gamma)\varepsilon))}{1-\gamma}\right)$ iterations.

4: $N \leftarrow O\left(\frac{\log(|\mathcal{S}| \cdot |\mathcal{W}|/\delta)}{(1-\gamma)^6 \varepsilon^2}\right)$ samples from the MDP generator for every $(\text{ctr}, W) \in \mathcal{S} \times \mathcal{W}$.

5: $M = \widetilde{O}\left(\frac{|\mathcal{S}| \cdot n \cdot k^5}{\varepsilon^3 (1-\gamma)^{12}} \log(1/\delta)\right)$ total samples from the valuation distribution $\mathcal{F}$, where $C$ is some absolute constant.

6: Draw $N$ samples from the generator of the MDP for every $(\text{ctr}, W) \in \mathcal{S} \times \mathcal{W}$.

7: Let $\widehat{P}_W(\text{ctr}|\text{ctr}) = \text{count}(\text{ctr}'|\text{ctr}, W)/N, \forall(\text{ctr}', \text{ctr}, W) \in \mathcal{S} \times \mathcal{S} \times \mathcal{W}$.

8: **for** $k = 1 \ldots K$ **do**

9:     Run the algorithm from Theorem B.2 using $M/(K \cdot |\mathcal{S}|)$ fresh samples from $\mathcal{F}$ to estimate $V^{(k)}(\text{ctr}) = \max_{a \in \mathcal{A}} R(\text{ctr}, a) + \gamma \sum_{\text{ctr}' \in \mathcal{S}} \mathbb{E}_{W \sim \mathcal{D}}[\widehat{P}_W(\text{ctr}'|\text{ctr})] V^{(k)}(\text{ctr}'), \forall \text{ctr} \in \mathcal{S}$.

10: **end for**

11: For every state $\text{ctr} \in \mathcal{S}$ return an $\varepsilon$-optimal solution to the problem $\widehat{\pi}(\text{ctr}) = \text{argmax}_{a \in \mathcal{A}} R(\text{ctr}, a) + \gamma \sum_{\text{ctr}' \in \mathcal{S}} \mathbb{E}_{W \sim \mathcal{D}^a}[\widehat{P}_W(\text{ctr}'|\text{ctr})] V^{(k)}(\text{ctr}')$ using the algorithm from Theorem B.2.

---

auction can be quite complex in general, even in the single-slot setting. In this section, we focus on designing a *simple* mechanism, for single slots, to achieve a constant-approximation to the optimal mechanism characterized in Section 3.1, following the spirit of the area of *"simple versus optimal mechanism"* [23]. In particular, our aim is to design a mechanism that is similar in spirit to a second-price auction with personalized reserves. The version of such an auction that is most relevant to this section works as follows. First we remove all advertisers whose bid is below their personalized reserve. Among the remaining bidders, we allocate to the highest bidder and charge them the larger of their reserve and the second-highest remaining bid. This is known as a second-price auction with eager reserves [34]. Note that this section focuses on the Bayesian Mechanism Design setting where the transition probability function $P_i(\cdot|\text{ctr})$, value function $V^*$, and value distributions $F_i$ are all known.

The main challenge of designing a good *simple* mechanism to achieve a constant-factor approximation to the optimal mechanism is that we need to take both the revenue guarantee in each round and the MDP transition into account. For example, if we have a *simple* auction that achieves a good revenue in round $t$ but leads to a bad transition for the user's click-through rate in the next round, then this may hurt the revenue dramatically in the long run. In an extreme case, the user may leave the platform and the seller loses all future revenue if our proposed simple mechanism incurs a terrible transition.

Given the above intuition, the key to our simple mechanism is to ensure that (i) we can obtain a constant approximation to

the optimal mechanism in each round and (ii) the transitions of our simple mechanism *exactly* matches the transition of the MDP induced by the optimal mechanism. In this section, we assume that each ad comes with a quality signal $q_i \in \{1, -1\}$ and that the transition functions depend only on the quality and not the actual ad shown. We refer to an ad with $q_i = 1$ as a good ad and an ad with $q_i = -1$ as a bad ad, although we do not require the assumption that showing a good ad is "better" than showing a bad ad. A formal description of the mechanism can be found in Mechanism 5 in Appendix C.1. Our main result in this section is the following theorem.

THEOREM 5.1. *Assume that the set of qualities has cardinality 2 and that the user state transitions depend only on the ad's quality. Given a policy $\pi^*$ with value function $V^*$, there is a policy $\pi$ that, for each state (i.e., in every round), the policy uses a two-stage second-price auction with personalized reserves and obtains an 8-approximation to the long-term discounted revenue obtained by $\pi^*$. In fact, $V^\pi(ctr) \geq \frac{1}{8}V^*(ctr)$ for all $ctr \in \mathcal{S}$.*

REMARK 5.2. *Let $S_{GOOD}$ (resp. $S_{BAD}$) be the set of "good" (resp. "bad") ads and $k = \min\{|S_{GOOD}|, |S_{BAD}|\}$. If $k \geq 2$ then our proof shows that $V^\pi(ctr) \geq \frac{1}{4}\left(1 - \frac{1}{k}\right) V^*(ctr)$ for all $ctr \in \mathcal{S}$. Thus, as the number of good and bad ads tend to infinity, we obtain a 4-approximation.*

Our key building block is the following mechanism.

LEMMA 5.3. *For any state of the MDP, there is a two-stage second-price mechanism $x$ with the following guarantee. Given a mechanism $x^*$ and a disjoint partition $S_1, S_2$ of $[n]$, the mechanism guarantees:*

*(1) The mechanism $x$ allocates to some bidder in $S_1$ with the same probability as $x^*$ and also for $S_2$. In other words $\sum_{i \in S_j} x_i = \sum_{i \in S_j} x_i^*$ for $j \in \{1, 2\}$.*

*(2) The revenue that $x$ obtains from $S_1$ is at least $1/4$ of the revenue that $x^*$ obtains from $S_1$.*

We now provide a high-level description on how to use the mechanism from Lemma 5.3 to prove Theorem 5.1; the details are in Appendix C.2. Given a state ctr of the MDP, we let $S_{\text{Good}} = \{i : q_i = 1\}$ be the set of good ads and $S_{\text{Bad}} = \{i : q_i = -1\}$ be the set of bad ads. We first compute the revenue contribution from $S_{\text{Good}}$ and $S_{\text{Bad}}$ according to $x^*$ for the current round. Let this be $R_{\text{Good}}$ and $R_{\text{Bad}}$, respectively. Assume that $R_{\text{Good}} \geq R_{\text{Bad}}$; the argument is analogous when $R_{\text{Bad}} \geq R_{\text{Good}}$. We set $S_1 = S_{\text{Good}}$ and $S_2 = S_{\text{Bad}}$ and run the mechanism from Lemma 5.3. The second guarantee from Lemma 5.3 ensures that the revenue that we obtain from $S_{\text{Good}}$ is at least $R_{\text{Good}}/4$. Since $R_{\text{Good}} \geq R_{\text{Bad}}$, it follows that we have an 8-approximation to the revenue that is obtained from $x^*$. Finally, note that the first guarantee of Lemma 5.3 and the fact that the transition depends only on the quality of the ad, it follows that we can exactly track the state transition of $x^*$.

We conclude this section by discussing the main ideas of the proof of Lemma 5.3. If $|S_1| = 1$ then in $x^*$, the sole bidder in $S_1$ is simply facing a reserve price that is determined by the other bidders. Thus, we offer this bidder the item at the same reserve price. The more interesting case is when $|S_1| \geq 2$. In the classical setting, it is known that the following mechanism yields a 2-approximation to the optimal revenue [18, 23, 34]. First, reject every bidder $i$ with

$\phi_i(v_i) < 0$ and then run a second-price auction among the remaining bidders. A naive extension would be to replace $\phi_i$ with the modified virtual value and 0 with the largest modified virtual value among bidders in $S_2$. However, this could allocate to a bidder whose (unmodified) virtual value is negative which is a technical challenge for the analysis. Thus, we add one additional step to increase the reserve of all but one of the bidders in $S_1$ to ensure that they are only allocated when their virtual value is non-negative. Since this decreases the probability that a bidder in $S_1$ wins, we compensate by decreasing the reserve of the remaining bidder. This turns out to make the analysis significantly simpler since the analysis can focus on the bidders with increased reserves. Finally, we allocate to an arbitrary bidder in $S_2$ (say, for free) with some probability to obtain the first guarantee of Lemma 5.3.

# 6 DISCUSSIONS AND FUTURE WORK

*Beyond binary types.* The characterization of the optimal MDP mechanism and our learning result in Section 3 and 4 do not require that the types are binary. On the other hand, we did make small use of the binary types in Section 5 in designing a simple two-stage mechanism. However, it is straightforward to extend the analysis to the case where the bidders can be partitioned into $k$ "categories" in which case we would achieve a $4k$-approximation. We leave it as another open question to see whether it is possible to obtain a $O(1)$-approximation, or even a $o(k)$-approximation.

*Better than 8-approximation for simple auctions.* We do not believe that the constant of 8 is tight in Theorem 5.1. In fact, as the number of good and bad advertisers grows, we are able to improve the constant arbitrary close to 4. Moreover, we note that when splitting it up into good and bad ads, we only focus on the revenue from one of the groups which incurs a factor 2 in the constant. Thus we conjecture that the constant can be made closer to 2.

*Dependence on state.* The mechanisms we described in this paper require knowledge of the user's state (i.e., its CTR). A natural question to ask is whether or not this is necessary. First, we note that a static mechanism has no hope of approximating the optimal MDP mechanism. Indeed, consider the following simple example. There are three CTR states $\{0, 1/2, 1\}$ and two advertisers. Advertiser 1 always has value $\varepsilon$ for some small $\varepsilon$ and is a good ad while advertiser 2 has value 1 and is a bad ad. The transitions are defined as follows. We assume 0 is an absorbing state. In states $1/2$ and 1, showing a good ad causes a deterministic transition to 1 while showing a bad ad causes a deterministic transition to 0 and $1/2$, respectively. The initial state is $1/2$. In this case, the optimal mechanism is to alternate between showing a good and bad ad to achieve per-round revenue of roughly $1/2$. On the other hand, any static mechanism can obtain per-round revenue of at most $\varepsilon$. However, this example does not necessarily rule out a dynamic, but stateless mechanism. For instance, one can consider using the past history of the shown ads and one can also observe "bandit-style" feedback to obtain some estimate of the user CTR. We leave this as yet another challenging open question to explore.

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

# Appendix

## A OMITTED PROOFS FROM SECTION 3

PROOF OF THEOREM 3.1. For a mechanism $a \in \mathcal{A}$, let $x^a$ denote the allocation of $a$ with the payment defined as discussed in Subsection 2.1. We begin with the Bellman equations which assert that

$$V^*(\text{ctr}) = \max_{a \in \mathcal{A}} \left\{ R(a, \text{ctr}) + \gamma \cdot \underset{\text{ctr}' \sim P(\cdot|a,\text{ctr})}{\mathbb{E}} \left[ V^*(\text{ctr}') \right] \right\}. \quad \text{(A.1)}$$

Let $X^a$ be the random set that gives the winning set of advertisers in the auction. Observe that

$$\underset{\text{ctr}' \sim P(\cdot|a,\text{ctr})}{\mathbb{E}} \left[ V^*(\text{ctr}') \right] = \mathbb{E} \left[ \underset{\text{ctr}'}{\mathbb{E}} \left[ V^*(\text{ctr}')|X^a \right] \right]$$

$$= \sum_{\substack{W \subseteq [n] \\ |W| \le k}} \underset{\text{ctr}'}{\mathbb{E}} \left[ V^*(\text{ctr}')|X^a = W \right] \cdot P(X^a = W)$$

$$= \sum_{\substack{W \subseteq [n] \\ |W| \le k}} \underset{\text{ctr}' \sim P_W(\cdot|\text{ctr})}{\mathbb{E}} \left[ V^*(\text{ctr}') \right] \cdot \underset{v}{\mathbb{E}} \left[ x^a_W(v) \right], \quad \text{(A.2)}$$

where in the last equality, we used that $P(X^a = W) = \mathbb{E}_v \left[ x^a_W(v) \right]$. Let $M_W(\text{ctr}) = \gamma \cdot \mathbb{E}_{\text{ctr}' \sim P_W(\cdot|\text{ctr})} \left[ V^*(\text{ctr}') \right]$. Plugging Eq. (A.2) into Eq. (A.1), we get that

$$V^*(\text{ctr}) = \max_{a \in A} \left\{ \sum_{\substack{W \subseteq [n] \\ 0 < |W| \le k}} \left[ \text{ctr} \cdot \underset{v}{\mathbb{E}} \left[ x^a_W(v) \sum_{i \in W} \phi_i(v_i) \right] \right. \right.$$

$$\left. + \underset{v}{\mathbb{E}} \left[ x^a_W(v) \cdot M_W(\text{ctr}) \right] + \underset{v}{\mathbb{E}} \left[ x^a_0(v) \cdot M_\emptyset(\text{ctr}) \right] \right\}$$

$$= \max_{a \in \mathcal{A}} \left\{ \sum_{\substack{W \subseteq [n] \\ 0 < |W| \le k}} \left[ \underset{v}{\mathbb{E}} \left[ x^a_W(v) \cdot \left( \text{ctr} \cdot \sum_{i \in W} \phi_i(v_i) \right. \right. \right. \right.$$

$$\left. \left. \left. \left. + M_W(\text{ctr}) - M_\emptyset(\text{ctr}) \right) \right] \right] \right\} + M_\emptyset(\text{ctr}). \quad \text{(A.3)}$$

In the first equality, we used that

$$R(a, \text{ctr}) = \sum_{\substack{W \subseteq [n] \\ 0 < |W| \le k}} \text{ctr} \cdot \underset{v}{\mathbb{E}} \left[ x^a_W(v) \cdot \sum_{i \in W} \phi_i(v_i) \right]$$

and also that $M_W(\text{ctr})$ is independent of the current values $v$ and so $\mathbb{E}_{\text{ctr}' \sim P_W(\cdot|\text{ctr})} \left[ V^*(\text{ctr}') \right] \cdot \mathbb{E}_v \left[ x^a_W(v) \right] = \mathbb{E}_v \left[ x^a_W(v) \cdot M_W(\text{ctr}) \right]$. In the second equality, we used that

$$x^a_\emptyset(v) = 1 - \sum_{\substack{W \subseteq [n] \\ 0 < |W| \le k}} x^a_W(v).$$

Finally, observe that we can optimize Eq. (A.3) by finding a set $W$ with $0 < |W| \le k$ that maximizes $\text{ctr} \cdot \sum_{i \in W} \phi_i(v_i) + M_W(\text{ctr}) - M_\emptyset(\text{ctr})$ if this quantity is positive. Otherwise, the mechanism does allocate to any advertiser. □

## B OMITTED PROOFS FROM SECTION 4

PROOF OF LEMMA 4.1. Let $V^\star$ be the optimal value function of the true MDP. Notice that $V^\star$ is independent of the randomness used in the estimation of $\widehat{P}$. We construct this MDP in the following

way: for every $(\text{ctr}, W, \text{ctr}') \in \mathcal{S} \times \mathcal{W} \times \mathcal{S}$, we set $\widehat{P}_W(\text{ctr}'|\text{ctr}) = \text{count}(\text{ctr}'|\text{ctr}, W)/N$, where $\text{count}(\text{ctr}'|\text{ctr}, W)$ is the number of transitions we observe in the samples from ctr to ctr' when the outcome is $W$. Thus, by using a Hoeffding bound and taking a union bound over $\mathcal{S} \times \mathcal{W}$ we see that with probability at least $1 - \delta$ it holds that for every $(\text{ctr}, W) \in \mathcal{S} \times \mathcal{W}$

$$|\langle P_W(\cdot|\text{ctr}) - \widehat{P}_W(\cdot|\text{ctr}), V^\star \rangle| \le \frac{1}{1 - \gamma} \cdot \sqrt{\frac{2 \log(2|\mathcal{S}||\mathcal{W}|/\delta)}{N}}.$$

Thus, with probability $1 - \delta$ we have that for every state $\text{ctr} \in \mathcal{S}$ and auction $a \in \mathcal{A}$ it holds

$$\left| \left\langle \underset{W \sim \mathcal{D}^a}{\mathbb{E}}[P_W(\cdot|\text{ctr})] - \underset{W \sim \mathcal{D}^a}{\mathbb{E}}[\widehat{P}_W(\cdot|\text{ctr})], V^\star \right\rangle \right| =$$

$$\left| \underset{W \sim \mathcal{D}^a}{\mathbb{E}} \left[ \left\langle P_W(\cdot|\text{ctr}) - \widehat{P}_W(\cdot|\text{ctr}), V^\star \right\rangle \right] \right| \le$$

$$\underset{W \sim \mathcal{D}^a}{\mathbb{E}} \left[ \left| \left\langle P_w(\cdot|\text{ctr}) - \widehat{P}_W(\cdot|\text{ctr}), V^\star \right\rangle \right| \right] \le$$

$$\frac{1}{1 - \gamma} \cdot \sqrt{\frac{2 \log(2|\mathcal{S}||\mathcal{W}|/\delta)}{N}}.$$

For the rest of the proof we condition on the previous event.

Let $\pi$ be some fixed (deterministic) policy. For every $(\text{ctr}, a) \in \mathcal{S} \times \mathcal{A}$ we define the operator $P^\pi$ as follows

$$P^\pi_{(\text{ctr},a),(\text{ctr}',a')} = \underset{W \sim \mathcal{D}^a}{\mathbb{E}}[P_W(\text{ctr}'|\text{ctr})] \quad \text{if } a' = \pi(\text{ctr}')$$

$$P^\pi_{(\text{ctr},a),(\text{ctr}',a')} = 0 \quad \text{otherwise} .$$

Similarly, we define the operator $\widehat{P}^\pi$

$$\widehat{P}^\pi_{(\text{ctr},a),(\text{ctr}',a')} = \underset{W \sim \mathcal{D}^a}{\mathbb{E}}[\widehat{P}_W(\text{ctr}'|\text{ctr})] \quad \text{if } a' = \pi(\text{ctr}')$$

$$\widehat{P}^\pi_{(\text{ctr},a),(\text{ctr}',a')} = 0 \quad \text{otherwise} .$$

Moreover, recall that $P_{(\text{ctr},a),\text{ctr}'} = \mathbb{E}_{W \sim \mathcal{D}^a}[\widehat{P}_W(\text{ctr}'|\text{ctr})], \forall (\text{ctr}, a) \in \mathcal{S} \times \mathcal{A}, \text{ctr}' \in \mathcal{S}$. Similarly for $\widehat{P}_{(\text{ctr},a),\text{ctr}'}$.

We start with the proof of the second inequality. Let $\mathcal{T}$ be the Bellman update rule w.r.t. the true MDP and $\widehat{\mathcal{T}}$ be the Bellman update rule w.r.t. the empirical MDP, i.e.,

$$\mathcal{T}(V)(\text{ctr}) = \max_{a \in \mathcal{A}} \left\{ R(\text{ctr}, a) + \left\langle \underset{W \sim \mathcal{D}^a}{\mathbb{E}}[P_W(\cdot|\text{ctr})], V \right\rangle \right\}$$

$$\mathcal{T}(Q)(\text{ctr}, a) = R(\text{ctr}, a) + \sum_{\text{ctr}' \in \mathcal{S}} \underset{W \sim \mathcal{D}^a}{\mathbb{E}}[P_W(\text{ctr}'|\text{ctr})] \cdot \max_{a \in \mathcal{A}} Q(\text{ctr}, a),$$

and similarly for $\widehat{\mathcal{T}}$. For the optimal $Q$-functions $Q^\star, \widehat{Q}^*$ of the true and the empirical MDP we have that

$$||Q^\star - \widehat{Q}^\star||_\infty = ||\mathcal{T}Q^\star - \widehat{\mathcal{T}}\widehat{Q}^\star||_\infty$$

$$\le ||\mathcal{T}Q^\star - R - \widehat{P}^{\pi^\star}Q^\star||_\infty + ||R + \widehat{P}^{\pi^\star}Q^\star - \widehat{\mathcal{T}}\widehat{Q}^\star||_\infty$$

$$= \gamma ||P^{\pi^\star}Q^\star - \widehat{P}^{\pi^\star}Q^\star||_\infty + \gamma ||\widehat{P}^{\pi^\star}Q^\star - \widehat{P}^{\widehat{\pi}^\star}\widehat{Q}^\star||_\infty$$

$$= \gamma ||(P - \widehat{P})V^\star||_\infty + \gamma ||\widehat{P}(V^\star - \widehat{V}^\star)||_\infty$$

$$\le \gamma ||(P - \widehat{P})V^\star||_\infty + \gamma ||V^\star - \widehat{V}^\star||_\infty$$

$$\le \gamma ||(P - \widehat{P})V^\star||_\infty + \gamma ||Q^\star - \widehat{Q}^\star||_\infty,$$

so solving for $||Q^\star - \widehat{Q}^\star||_\infty$ gives

$$||Q^\star - \widehat{Q}^\star||_\infty \leq \frac{\gamma}{1-\gamma}||(P - \widehat{P})V^\star||_\infty$$

$$\leq \frac{\gamma}{(1-\gamma)^2} \cdot \sqrt{\frac{2\log(2|\mathcal{S}||\mathcal{W}|/\delta)}{N}}.$$

Finally, to get the desired inequality notice that

$$||V^\star - \widehat{V}^\star||_\infty \leq ||Q^\star - \widehat{Q}^\star||_\infty.$$

We now shift our attention to the first inequality.

$$Q^\pi(\mathrm{ctr}, a) = R(\mathrm{ctr}, a) + \gamma \langle \underset{W\sim\mathcal{D}^a}{\mathbb{E}}[P_W(\cdot|\mathrm{ctr})], V^\pi \rangle \implies$$

$$Q^\pi = R + \gamma P^\pi Q^\pi \implies$$

$$Q^\pi = (I - \gamma P^\pi)^{-1}R,$$

where the last step follows by the fact that $(I - \gamma P^\pi)$ is a bounded linear operator on the vector space $\mathcal{S} \times \mathcal{A}$ with respect to the sup-norm and the Neumann series converges in the operator norm, so the inverse of the operator exists and is given by the formula $(I - \gamma P^\pi)^{-1} = \sum_{\gamma=0}^{\infty} \gamma^i(P^\pi)^i$. Similarly, we have

$$\widehat{Q}^\pi = (I - \gamma \widehat{P}^\pi)^{-1}R.$$

Equipped with the above notation, we can write

$$Q^\pi - \widehat{Q}^\pi = (I - \gamma P^\pi)^{-1}R - (I - \gamma \widehat{P}^\pi)^{-1}R$$

$$= (I - \gamma P^\pi)^{-1}\left((I - \gamma\widehat{P}^\pi) - (I - \gamma P^\pi)\right)\widehat{Q}^\pi$$

$$= \gamma(I - \gamma P^\pi)^{-1}(P^\pi - \widehat{P}^\pi)\widehat{Q}^\pi$$

$$= \gamma(I - \gamma P^\pi)^{-1}(P - \widehat{P})\widehat{V}^\pi.$$

We can write

$$(I - \gamma P^\pi)^{-1} = \sum_{\gamma=0}^{\infty} \gamma^i(P^\pi)^i,$$

where $||(P^\pi)^i||_\infty = 1$. Thus, we have

$$||\gamma(I - \gamma P^\pi)^{-1}(P - \widehat{P})\widehat{V}^\pi||_\infty \leq \gamma \sum_{i=0}^{\infty}||\gamma^i(P^\pi)^i(P - \widehat{P})\widehat{V}^\pi||_\infty$$

$$\leq \gamma \sum_{i=0}^{\infty}\gamma^i||(P - \widehat{P})\widehat{V}^\pi||_\infty$$

$$\leq \frac{\gamma}{(1-\gamma)}||(P - \widehat{P})\widehat{V}^\pi||_\infty,.$$

We now focus on the term $||(P - \widehat{P})\widehat{V}^\pi||_\infty$. We have that

$$||(P - \widehat{P})\widehat{V}^\pi||_\infty \leq$$

$$||(P - \widehat{P})V^\star||_\infty + ||(P - \widehat{P})(\widehat{V}^\pi - V^\star)||_\infty \leq$$

$$||(P - \widehat{P})V^\star||_\infty + 2||\widehat{V}^\pi - V^\star||_\infty \leq$$

$$||(P - \widehat{P})V^\star||_\infty + 2||\widehat{V}^\pi - \widehat{V}^\star||_\infty + 2||\widehat{V}^\star - V^\star||_\infty \leq$$

$$||(P - \widehat{P})V^\star||_\infty + 2||\widehat{V}^\pi - \widehat{V}^\star||_\infty + 2||\widehat{Q}^\star - Q^\star||_\infty \leq$$

$$\frac{1}{1-\gamma} \cdot \sqrt{\frac{2\log(2|\mathcal{S}||\mathcal{W}|/\delta)}{N}} + 2\varepsilon_{\mathrm{opt}}+$$

$$2\frac{\gamma}{(1-\gamma)^2} \cdot \sqrt{\frac{2\log(2|\mathcal{S}||\mathcal{W}|/\delta)}{N}} \leq$$

$$2\varepsilon_{\mathrm{opt}} + 3\frac{\gamma}{(1-\gamma)^2} \cdot \sqrt{\frac{2\log(2|\mathcal{S}||\mathcal{W}|/\delta)}{N}},$$

where $\varepsilon_{\mathrm{opt}} = ||\widehat{V}^\pi - \widehat{V}^\star||_\infty$. Combining the above inequalities, we get that

$$||Q^\pi - \widehat{Q}^\pi||_\infty \leq \frac{2\gamma\varepsilon_{\mathrm{opt}}}{1-\gamma} + 3\frac{\gamma^2}{(1-\gamma)^3} \cdot \sqrt{\frac{2\log(2|\mathcal{S}||\mathcal{W}|/\delta)}{N}}.$$

Finally, applying the inequality

$$||V^\pi - \widehat{V}^\pi||_\infty \leq ||Q^\pi - \widehat{Q}^\pi||_\infty,$$

gives the desired result. $\square$

PROOF OF THEOREM 4.2. Let

$$L_{V\widetilde{\pi}}(\mathrm{ctr}) = V^\star(\mathrm{ctr}) - V^{\widetilde{\pi}}(\mathrm{ctr})$$

$$\mathrm{ctr}^\star = \underset{\mathrm{ctr}\in\mathcal{S}}{\mathrm{argmax}}\, L_{V\widetilde{\pi}}(\mathrm{ctr}).$$

Consider the state $\mathrm{ctr}^\star \in \mathcal{S}$ and let $a = \pi^\star(\mathrm{ctr}^\star), b = \widetilde{\pi}(\mathrm{ctr}^\star)$. By definition, we have that

$$R(\mathrm{ctr}^\star, b) + \gamma \sum_{\mathrm{ctr}'} \underset{W\sim\mathcal{D}^b}{\mathbb{E}}[P_W(\mathrm{ctr}'|\mathrm{ctr}^\star)] \cdot \widetilde{V}(\mathrm{ctr}') \geq$$

$$R(\mathrm{ctr}^\star, a) + \gamma \sum_{\mathrm{ctr}'} \underset{W\sim\mathcal{D}^a}{\mathbb{E}}[P_W(\mathrm{ctr}'|\mathrm{ctr}^\star)] \cdot \widetilde{V}(\mathrm{ctr}') - \varepsilon' \implies$$

$$R(\mathrm{ctr}^\star, b) + \gamma \sum_{\mathrm{ctr}'} \underset{W\sim\mathcal{D}^b}{\mathbb{E}}[P_W(\mathrm{ctr}'|\mathrm{ctr}^\star)] \cdot (V^\star(\mathrm{ctr}') + \varepsilon_{\mathrm{opt}}) \geq$$

$$R(\mathrm{ctr}^\star, a) + \gamma \sum_{\mathrm{ctr}'} \underset{W\sim\mathcal{D}^a}{\mathbb{E}}[P_W(\mathrm{ctr}'|\mathrm{ctr}^\star)] \cdot (V^\star(\mathrm{ctr}') - \varepsilon_{\mathrm{opt}}) - \varepsilon' \implies$$

$$R(\mathrm{ctr}^\star, b) + \gamma \sum_{\mathrm{ctr}'} \underset{W\sim\mathcal{D}^b}{\mathbb{E}}[P_W(\mathrm{ctr}'|\mathrm{ctr}^\star)] \cdot V^\star(\mathrm{ctr}') \geq$$

$$R(\mathrm{ctr}^\star, a) + \gamma \sum_{\mathrm{ctr}'} \underset{W\sim\mathcal{D}^a}{\mathbb{E}}[P_W(\mathrm{ctr}'|\mathrm{ctr}^\star)] \cdot V^\star(\mathrm{ctr}') - 2\gamma\varepsilon_{\mathrm{opt}} - \varepsilon' \implies$$

$$R(\mathrm{ctr}^\star, a) - R(\mathrm{ctr}^\star, b) \leq$$

$$2\gamma\varepsilon_{\mathrm{opt}} + \varepsilon' + \gamma \sum_{\mathrm{ctr}'\in\mathcal{S}} (\underset{W\sim\mathcal{D}^b}{\mathbb{E}}[P_W(\mathrm{ctr}'|\mathrm{ctr}^\star)]-$$

$$\underset{W\sim\mathcal{D}^a}{\mathbb{E}}[P_W(\mathrm{ctr}'|\mathrm{ctr}^\star)])V^\star(\mathrm{ctr}'),$$

where the first derivation follows from the property of $\widetilde{V}$ and the remaining ones by rearranging the terms. By definition, we have

that

$$L_{V^{\widetilde{\pi}}}(\text{ctr}^*) = V^\star(\text{ctr}^\star) - V^{\widetilde{\pi}}(\text{ctr}^\star)$$

$$= R(\text{ctr}^\star, a) - R(\text{ctr}^\star, b) +$$

$$\gamma \sum_{\text{ctr}' \in \mathcal{S}} (\underset{W \sim \mathcal{D}^a}{\mathbb{E}}[P_W(\text{ctr}'|\text{ctr}^\star)]V^\star(\text{ctr}') -$$

$$\underset{W \sim \mathcal{D}^b}{\mathbb{E}}[P_W(\text{ctr}'|\text{ctr}^\star)]V^{\widetilde{\pi}}(\text{ctr}'))$$

Substituting the previous inequality to this one we get

$$L_{V^{\widetilde{\pi}}}(\text{ctr}^*) \leq 2\gamma\varepsilon_{\text{opt}} + \varepsilon' + \gamma \sum_{\text{ctr}' \in \mathcal{S}} (\underset{W \sim \mathcal{D}^b}{\mathbb{E}}[P_W(\text{ctr}'|\text{ctr}^\star)]V^\star(\text{ctr}') -$$

$$\underset{W \sim \mathcal{D}^a}{\mathbb{E}}[P_W(\text{ctr}'|\text{ctr}^\star)]V^\star(\text{ctr}') + \underset{W \sim \mathcal{D}^a}{\mathbb{E}}[P_W(\text{ctr}'|\text{ctr}^\star)]V^\star(\text{ctr}') -$$

$$\underset{W \sim \mathcal{D}^b}{\mathbb{E}}[P_W(\text{ctr}'|\text{ctr}^\star)]V^{\widetilde{\pi}}(\text{ctr}')) \implies$$

$$L_{V^{\widetilde{\pi}}}(\text{ctr}^*) \leq 2\gamma\varepsilon_{\text{opt}} + \varepsilon' +$$

$$\gamma \sum_{\text{ctr}' \in \mathcal{S}} \underset{W \sim \mathcal{D}^b}{\mathbb{E}}[P_W(\text{ctr}'|\text{ctr}^\star)](V^\star(\text{ctr}') - V^{\widetilde{\pi}}(\text{ctr}')).$$

By definition, $V^\star(\text{ctr}') - V^{\widetilde{\pi}}(\text{ctr}') \leq L_{V^{\widetilde{\pi}}}(\text{ctr}^*)$. Thus, we get

$$L_{V^{\widetilde{\pi}}}(\text{ctr}^*) \leq 2\gamma\varepsilon_{\text{opt}} + \varepsilon' + \gamma L_{V^{\widetilde{\pi}}}(\text{ctr}^*),$$

so the result follows by solving for $L_{V^{\widetilde{\pi}}}(\text{ctr}^*)$. □

PROOF OF THEOREM 4.3. Notice that

$$||V^{(k)} - V^\star||_\infty \leq ||\mathcal{T}V^{(k-1)} - \mathcal{T}V^\star||_\infty + ||\varepsilon_{k-1}||_\infty$$

$$\leq \gamma||V^{(k-1)} - V^\star||_\infty + \varepsilon.$$

Thus, applying the inequality recursively we see that

$$||V^{(k)} - V^\star||_\infty \leq \gamma^k||V^\star||_\infty + \sum_{k=1}^K \gamma^{k-i}\varepsilon$$

$$\leq \gamma^k||V^\star||_\infty + \frac{\varepsilon}{1-\gamma}.$$

In order to prove the desired bound we use Theorem 4.2 and we get

$$||V^{\pi_k} - V^\star||_\infty \leq \frac{2\gamma||V^{(k)} - V^\star||_\infty + \varepsilon'}{1-\gamma}$$

$$\leq \frac{2\gamma}{1-\gamma} \cdot \left(\gamma^k||V^\star||_\infty + \frac{\varepsilon}{1-\gamma} + \frac{\varepsilon'}{2\gamma}\right)$$

$$\leq \frac{2\gamma}{(1-\gamma)^2} \cdot \left(\gamma^k + \varepsilon + \frac{(1-\gamma)\varepsilon'}{2\gamma}\right).$$

□

PROOF OF THEOREM 4.4. From Lemma 4.1 we know that constructing the emprical MDP using $N = O\left(\frac{\log(|\mathcal{S}| \cdot |\mathcal{W}|/\delta)}{(1-\gamma)^6\varepsilon^2}\right)$ samples from every state-output pair we have that, with probability at least $1 - \delta$, $||V^\star - \widehat{V}^\star||_\infty \leq \varepsilon/3$ and $||V^\pi - \widehat{V}^\pi|| \leq 2\gamma\varepsilon_{\text{opt}}/(1-\gamma) + \varepsilon/6$, for any policy $\pi$, where $\varepsilon_{\text{opt}} = ||\widehat{V}^\pi - \widehat{V}^\star||_\infty$. In the empirical MDP $\widehat{P}$ we can run exact value iteration (i.e., solve with $\varepsilon_k = 0$) for $O\left(\frac{\ln(1/(\varepsilon(1-\gamma)))}{1-\gamma}\right)$ iterations and using the guarantees of Theorem 4.3 we estimate a policy $\pi_k$ such that $||\widehat{V}^{\pi_k} - \widehat{V}^\star||_\infty \leq$

$(1-\gamma)\varepsilon/12$. Thus, for this policy we have that $||\widehat{V}^{\pi_k} - V^{\pi_k}||_\infty \leq \varepsilon/3$. Combining these three inequalities, we have that

$$||V^{\pi_k} - V^\star||_\infty \leq ||V^{\pi_k} - \widehat{V}^{\pi_k}||_\infty + ||\widehat{V}^{\pi_k} - \widehat{V}^\star||_\infty + ||\widehat{V}^\star - V^\star||_\infty$$

$$\leq \varepsilon. \qquad \square$$

LEMMA B.1 ([15]). *Let* $f : \mathcal{X}^n \to [0, 1]$, *where* $\mathcal{X}$ *is some measurable set. Let* $\{F_i\}_{i \in [n]}$ *be distributions on* $\mathcal{X}$, *and let* $\mathcal{F} = F_1 \times \ldots \times F_n$. *Suppose* $\mu = \mathbb{E}_{(v_1,\ldots,v_n) \sim \mathcal{F}}[f(v_1, \ldots, v_n)]$. *Let* $v_{i_1}, \ldots, v_{i_m}$ *be m i.i.d. samples from* $F_i$. *Let* $U_i$ *be the uniform distribution over* $\{v_{i_1}, \ldots, v_{i_m}\}$ *and* $U = U_1 \times \ldots \times U_n$. *Then, we have*

$$\underset{v_{i_j} \sim F_i, \forall i \in [n], j \in [m]}{\mathbf{Pr}} \left[ \left| \underset{(\widehat{v}_1,\ldots,\widehat{v}_n) \sim U}{\mathbb{E}} f(\widehat{v}_1, \ldots, \widehat{v}_n) - \mu \right| \geq 2\varepsilon \right]$$

$$\leq 2e^{\frac{-2m\varepsilon^2}{4\mu+\varepsilon} - \ln(\varepsilon)}.$$

PROOF OF LEMMA 4.6. In order to prove the result we show that given the optimal mechanism $M$ for $\mathcal{F}$ we can construct a mechanism $\widehat{M}$ that achieves revenue on $\widehat{\mathcal{F}}$ that is at most $k \cdot \varepsilon$ worse. From Theorem 3.1 we know that for every input $v_1, \ldots, v_n$, $M$ allocates the slots to a set of bidders $W, 0 \leq |W| \leq k$, that maximize

$$\sum_{i \in W} \phi_i(v_i) + \gamma \left( \underset{\text{ctr}' \sim P_W(\cdot|\text{ctr})}{\mathbb{E}} [V^*(\text{ctr}')] - \underset{\text{ctr}' \sim P_\emptyset(\cdot|\text{ctr})}{\mathbb{E}} [V^*(\text{ctr}')] \right),$$

where $\phi_i$ if the (ironed) virtual value of bidder $i$. We assume without loss of generality that $M$ breaks ties deterministically among sets of bidders that maximize this quantity. We define the allocation rule of $\widehat{M}$ for the distribution $\widehat{\mathcal{F}}$ as follows.

- Let $\widehat{v} = (\widehat{v}_1, \ldots, \widehat{v}_n) \in \{0, \varepsilon, \ldots, h\}^n$ be the reported value profile.
- For every bidder $i \in [n]$ sample $v_i$ from $F_i$ conditioned on $\widehat{v}_i \leq v_i < \widehat{v}_i + \varepsilon$.
- Use the allocation rule of $M$ on $(v_1, \ldots, v_n)$.

We first show that there is a payment rule that makes $\widehat{M}$ truthful. Suppose that we fix the (internal) randomness used in the second step. Then, for every bidder $i \in [n]$, holding the bids of the rest fixed, the allocation rule is a step function. Therefore, we can make the rule IC by charging the winners the minimum bid they would need to submit to be part of the winning set. Since the mechanism is IC for every realization of its random bits, it is IC overall.

The next step is to compare the expected modified revenue of $\widehat{M}$ w.r.t. $\widehat{\mathcal{F}}$ to the expected modified revenue of $M$ w.r.t. $F$. In order to do so, we couple two random variables, the modified revenue of $M$ over a sample $v = (v_1, \ldots, v_n) \sim \mathcal{F}$ and the modified revenue of $\widehat{M}$ over a sample $\widehat{v} = (\widehat{v}_1, \ldots, \widehat{v}_n) \sim \widehat{\mathcal{F}}$. To be more precise, we will couple valuation profiles $v, \widehat{v}$ so that $v$ is the output of the random redraw (i.e., step 2 of $\widehat{M}$.). For each such pair, by definition of $\widehat{M}$, the allocation is the same as in $M$. Therefore, the contribution to the modified revenue of the term $\gamma \left( \mathbb{E}_{\text{ctr}' \sim P_W(\cdot|\text{ctr})} [V^*(\text{ctr}')] - \mathbb{E}_{\text{ctr}' \sim P_\emptyset(\cdot|\text{ctr})} [V^*(\text{ctr}')] \right)$ is the same across the two mechanisms. Let us now focus on the payments. Suppose that $W^\star$ is the winning set. By definition, the winning threshold of every $i \in W^\star$ in $\widehat{M}$ is obtained by rounding down the winning threshold of $i$ in $M$ to closest multiple of $\varepsilon$. Thus, the payments of every bidder are at most $\varepsilon$ worse. Hence, we see that

the expected modified revenue of $\widehat{M}$ is at most $k \cdot \varepsilon$ worse than the expected modified revenue of $M$. □

PROOF OF LEMMA 4.8. Let $M^\star$ be the optimal mechanism, $v = (v_1, \ldots, v_n)$ be some valuation profile and let $W^\star$ be the set of bidders that $\mathcal{M}^\star$ allocates the slots to under input $v$. Consider threshold auction that rounds down the ironed virtual value $\phi_i(v_i)$ of each bidder to the closest multiple of $\varepsilon/k$ in $\{-\infty\} \cup \left\{ \frac{-2k}{1-\gamma}, \frac{-2k}{1-\gamma} + \varepsilon/k, \ldots, 1 \right\}$. Let $\widetilde{\phi}_i(v_i)$ denote the rounded virtual value. It is not hard to see that if $\widetilde{\phi}_i(v_i) = -\infty$ then $i \notin W^*$, since $\phi_i(v_i) < 2k/(1-\gamma)$. Assume that $i \in W^*$. Then $\sum_{j \in W^\star \setminus \{i\}} \phi_j(v_j) \le k$ and

$$\gamma \left( \underset{\text{ctr}' \sim P_W(\cdot|\text{ctr})}{\mathbb{E}} \left[ V(\text{ctr}') \right] - \underset{\text{ctr}' \sim P_\emptyset(\cdot|\text{ctr})}{\mathbb{E}} \left[ V(\text{ctr}') \right] \right) \le \frac{k}{1-\gamma},$$

so the modified virtual value of $W^\star$ is negative. Let $\widehat{W}^\star$ be the set of bidders selected by $M$. Then, we have that

$$\text{ctr} \cdot \sum_{i \in \widehat{W}^\star} \widetilde{\phi}_i(v_i)$$

$$+ \gamma \left( \underset{\text{ctr}' \sim P_{\widehat{W}^\star}(\cdot|\text{ctr})}{\mathbb{E}} \left[ V(\text{ctr}') \right] - \underset{\text{ctr}' \sim P_\emptyset(\cdot|\text{ctr})}{\mathbb{E}} \left[ V(\text{ctr}') \right] \right) \ge$$

$$\text{ctr} \cdot \sum_{i \in W^\star} \widetilde{\phi}_i(v_i)$$

$$+ \gamma \left( \underset{\text{ctr}' \sim P_{W^\star}(\cdot|\text{ctr})}{\mathbb{E}} \left[ V(\text{ctr}') \right] - \underset{\text{ctr}' \sim P_\emptyset(\cdot|\text{ctr})}{\mathbb{E}} \left[ V(\text{ctr}') \right] \right) \ge$$

$$\text{ctr} \cdot \sum_{i \in W^\star} \phi_i(v_i)$$

$$+ \gamma \left( \underset{\text{ctr}' \sim P_{W^\star}(\cdot|\text{ctr})}{\mathbb{E}} \left[ V(\text{ctr}') \right] - \underset{\text{ctr}' \sim P_\emptyset(\cdot|\text{ctr})}{\mathbb{E}} \left[ V(\text{ctr}') \right] \right) - k \cdot (\varepsilon/k).$$

Thus, for every bid profile $v$ the modified virtual welfare of $M$ is at most $\varepsilon$ worse than the modified virtual welfare of the optimal mechanism. The result follows by integrating over all the valuation profiles. □

PROOF OF LEMMA 4.9. Based on Lemma 4.8 we know that there is a threshold auction at resolution $\varepsilon/(2k)$ whose modified revenue is at most $\varepsilon/2$ worse than the optimal one. Since any tie-breaking rule gives the same revenue in expectation, we can assume w.l.o.g. that it is deterministic. For every bidder $i \in [n]$ and every valuation $v_{ij} \in B$ such threshold auctions have $\frac{k(4k+1-\gamma)}{\varepsilon(1-\gamma)} + 1$ possible mappings for $v_{ij}$. Thus, since every threshold auction can be described by such a mapping there are at most

$$L \le \left( \frac{k(4k+1-\gamma)}{\varepsilon(1-\gamma)} + 1 \right)^{n|B|},$$

different such mappings. For every such mapping $\sigma$ a mechanism $M_\sigma$[9] treats $\sigma$ as the virtual value function and then allocates the slots to the set of at most $k$ bidders that maximize the modified virtual welfare. Since the values are bounded by 1 and there are $k$ slots, we know that the modified revenue is at most $k/(1-\gamma)$. Let $f_\sigma$ be the modified revenue of $M_\sigma$. Then, we use Lemma B.1 on every $f_\sigma$ by scaling them appropriately so that the modified

---

[9]In order for this to be a truthful mechanism we need the mapping to be monotone, but we only need an upper bound on the number of such mechanisms.

revenue is in $[0, 1]$, and we set the approximation parameter to be $\widetilde{\varepsilon} = (1-\gamma)\varepsilon/(4k^2)$ by taking

$$m = \widetilde{O} \left( \frac{n \cdot |B| \cdot k^4}{\varepsilon^2 (1-\gamma)^2} \log(1/\delta) \right).$$

Let $\widetilde{\mu}$ be an upper bound on the expected modified revenue of $M_\sigma$. We denote by $\widetilde{\text{Rev}}(M_\sigma, \mathcal{F})$ the modified revenue of $M_\sigma$ with respect to $\mathcal{F}$ and by $\widetilde{\text{Rev}}(M_\sigma, \widehat{\mathcal{F}})$ the modified revenue of $M_\sigma$ with respect to $\widehat{\mathcal{F}}$ Hence, for each such function we have

$$\Pr[|\widetilde{\text{Rev}}(M_\sigma, \mathcal{F}) - \widetilde{\text{Rev}}(M_\sigma, \widehat{\mathcal{F}})| \ge \varepsilon/(4k)]$$

$$\le 2e^{\frac{2m\widetilde{\varepsilon}^2}{4\widetilde{\mu}+\varepsilon} - \ln(\widetilde{\varepsilon})}$$

$$\le \delta/L.$$

Thus, by taking a union bound over all $M_\sigma$ we can see that with probability at least $1 - \delta$ for any $\sigma$ it holds that

$$|\widetilde{\text{Rev}}(M_\sigma, \mathcal{F}) - \widetilde{\text{Rev}}(M_\sigma, \widehat{\mathcal{F}})| \le \varepsilon/(4k).$$

Notice that it is not computationally efficient to output the best threshold mechanism with respect to the empirical distribution. Thus, we take the following approach. First, we compute the (ironed) virtual values $\widetilde{\phi}_i$ w.r.t. the empirical distribution $\widehat{\mathcal{F}}, \forall i \in [n]$. This can be done in polynomial time in the number of samples $m$ using the result of [17]. Then, we output the mechanism $\widehat{M}$ which, for every input profile $v$ rounds the virtual values (w.r.t. $\mathcal{F}$) to the closest multiple of $\varepsilon/(4k)$ and selects the set $W$ that maximizes the modified virtual welfare. This mechanism has modified revenue w.r.t. the empirical distribution $\widehat{\mathcal{F}}$ at most $\varepsilon/4$ worse than the optimal w.r.t. $\widehat{\mathcal{F}}$. Thus, it also has modified revenue at most $\varepsilon/4$ worse than the optimal threshold mechanism w.r.t. $\widehat{\mathcal{F}}$. Thus, the modified revenue of $\widehat{M}$ w.r.t. $\mathcal{F}$ is at most $\varepsilon/2$ worse than that of the optimal threshold mechanism w.r.t. the true distribution $\mathcal{F}$, which in turn is at most $\varepsilon/2$ worse than the optimal modified revenue w.r.t. $\mathcal{F}$. Hence, we see the modified revenue of $\widehat{M}$ w.r.t. $\mathcal{F}$ is at most $\varepsilon$ worse than the optimal. □

---

**Algorithm 2** Approximately Optimal Modified Revenue Mechanism from Samples

1: **Input**: $m$ i.i.d. samples from $\mathcal{F} = F_1 \times \ldots \times F_n$, where each $F_i$ is supported on $[0, 1]$, discount factor $\gamma$, number of slots $k$, value function $V : \mathcal{S} \to [0, k/(1-\gamma)]$, transition kernel of the MDP $P$, approximation parameter $\varepsilon$.
2: **Output**: A truthful mechanism.
3: Round all the samples down to the closest multiple of $\varepsilon/(2k)$.
4: Let $\widehat{\mathcal{F}} = \widehat{F}_1 \times \ldots \times \widehat{F}_n$ be the empirical distribution on the rounded $m$ samples
5: $\widehat{\phi}_i \leftarrow$ ironed virtual value of bidder $i$ w.r.t. $\widehat{F}, \forall i \in [n]$
6: $M \leftarrow$ output of Algorithm 3 with inputs $\{\widehat{\phi}_i\}_{i \in [n]}, V, \gamma, P$, and resolution $\beta = \varepsilon/(4k)$.
7: $\widehat{M} \leftarrow$ the mechanism that rounds down all of its inputs to multiples of $\varepsilon/(2k)$ and then runs $M$
8: $\widehat{\text{OPT}}(\mathcal{F}) \leftarrow$ revenue of $\widehat{M}$ w.r.t. $\widehat{\mathcal{F}}$.
9: Return $\widehat{M}, \widehat{\text{OPT}}(\mathcal{F})$.

---

**Algorithm 3** Rounded Virtual Value Mechanism

1: **Input**: Ironed virtual value functions $\{\widehat{\phi}_i\}_{i\in[n]}$, discount factor $\gamma$, number of slots $k$, value function $V : \mathcal{S} \to [0, k/(1-\gamma)]$, transition kernel of the MDP $P$, resolution parameter $\beta$, valuation profile $v = (v_1, \ldots, v_n)$.

2: **Output**: Allocation of the slots.

3: $\widehat{v}_i \leftarrow$ closest rounded down value of $\widehat{\phi}_i(v_i)$ in $\{-\infty\} \cup \left\{\frac{-2k}{1-\gamma}, \frac{-2k}{1-\gamma} + \beta, \ldots, 1\right\}$.

4: $R \leftarrow \max_{0<|W|\le k} \mathrm{ctr} \cdot \sum_{i\in W} \widehat{v}_i + \gamma(\mathbb{E}_{\mathrm{ctr}'\sim P_W(\cdot|\mathrm{ctr})}[V(\mathrm{ctr}')] - \mathbb{E}_{\mathrm{ctr}'\sim P_\emptyset(\cdot|\mathrm{ctr})}[V(\mathrm{ctr}')])$.

5: $W^\star \leftarrow \operatorname{argmax}_{0<|W|\le k} \mathrm{ctr} \cdot \sum_{i\in W} \widehat{v}_i + \gamma(\mathbb{E}_{\mathrm{ctr}'\sim P_W(\cdot|\mathrm{ctr})}[V(\mathrm{ctr}')] - \mathbb{E}_{\mathrm{ctr}'\sim P_\emptyset(\cdot|\mathrm{ctr})}[V(\mathrm{ctr}')])$.

6: **if** $R < 0$ **then**

7:     Return $\emptyset$.

8: **else**

9:     Return $W^\star$.

10: **end if**

---

THEOREM B.2. *Given any product value distribution $\mathcal{F} = F_1 \times \ldots \times F_n$ where every $F_i$ is supported on $[0,1]$, and any $\varepsilon > 0, \delta > 0$, Algorithm 2 takes as input $m$ i.i.d. samples from $\mathcal{F}$ and outputs an auction $M$ that achieves modified revenue at least $\widehat{\mathrm{OPT}}(\mathcal{F}) - \varepsilon$, with probability at least $1 - \delta$, whenever*

$$m = \widetilde{O}\left(\frac{n \cdot k^5}{\varepsilon^3(1-\gamma)^2}\log(1/\delta)\right),$$

*as well as an estimate $\widetilde{\mathrm{OPT}}(\mathcal{F})$ with the property $|\widehat{\mathrm{OPT}}(\mathcal{F}) - \widetilde{\mathrm{OPT}}(\mathcal{F})| \le \varepsilon$. Moreover, the running time of the algorithm to output $M$ is $\mathrm{poly}(m)$ and the running time to output the estimate of the modified revenue is $\mathrm{poly}(m, n^k)$.*

PROOF OF THEOREM B.2. This is a corollary of Lemma 4.6 and Lemma 4.9. By doing a discretization in the valuation distribution with parameter $\varepsilon/2k$ we know that the modified revenue decreases by at most $\varepsilon/2$. Then, we can use Lemma 4.9 with parameter $\varepsilon/2$, so overall the modified revenue loss is at most $\varepsilon$. In order to output the estimate of the revenue, we need to run the mechanism on the empirical distribution we have constructed, which takes time $\mathrm{poly}(m, n^k)$. □

PROOF OF THEOREM 4.10. This result is, essentially, a corollary of various results we have shown so far. First, notice that by taking $O\left(\frac{\log(|\mathcal{S}|\cdot|\mathcal{W}|/\delta)}{(1-\gamma)^6\varepsilon^2}\right)$ from every state-outcome pair $(\mathrm{ctr}, W) \in \mathcal{S} \times \mathcal{W}$ and constructing the empirical MDP $\widehat{P}$ guarantees that (cf. Lemma 4.1), with probability at least $1 - \delta/2$, for any policy $\pi$

$$||V^\pi - \widehat{V}^\pi||_\infty \le \frac{2\gamma\varepsilon_{\mathrm{opt}}}{1-\gamma} + \frac{\varepsilon}{6}$$

$$||V^\star - \widehat{V}^\star||_\infty \le \frac{\varepsilon}{3},$$

where $\varepsilon_{\mathrm{opt}} = ||\widehat{V}^\pi - \widehat{V}^\star||_\infty$. We call this event $\mathcal{E}_0$ and we condition on it for the rest of the proof. Thus, it suffices to compute a policy $\pi$ for which $||\widehat{V}^\pi - \widehat{V}^\star||_\infty = O((1-\gamma)\varepsilon)$. To that end, Theorem 4.3 shows that if run value iteration for $k$ rounds in the empirical MDP, estimate every update with accuracy $\varepsilon'$, and define $\pi_k$ to be

an $\varepsilon'$-greedy policy with respect to $\widehat{V}^{(k)}$, then $||\widehat{V}^{\pi_k} - \widehat{V}^\star||_\infty \le \frac{6\gamma}{(1-\gamma)^2} \cdot (\gamma^k + \varepsilon')$. Thus, by choosing $\varepsilon' = O((1-\gamma)^3\varepsilon), K = O\left(\frac{\log(1/((1-\gamma)\varepsilon))}{1-\gamma}\right)$ we end up with $||\widehat{V}^{\pi_K} - \widehat{V}^\star||_\infty \le (1-\gamma)\varepsilon/12$. Hence, plugging in the guarantees of Theorem B.2 with accuracy $\varepsilon' = O((1-\gamma)^3\varepsilon)$, and confidence $\delta/(2k|\mathcal{S}|)$ we see that that with probability at least $1 - \delta/2$ all the calls will return an $\varepsilon'$-optimal auction as well as their (expected) revenue with accuracy $\varepsilon'$. The number of samples from $\mathcal{F}$ needed for each iteration and each state is

$$m = \widetilde{O}\left(\frac{n \cdot k^5}{\varepsilon^3(1-\gamma)^{11}}\log(|\mathcal{S}|/\delta)\right).$$

We call this event $\mathcal{E}_1$ and we condition on it for the rest of the proof. The total number of samples we draw from $\mathcal{F} = F_1 \times \ldots \times F_n$ is

$$O(K \cdot |\mathcal{S}| \cdot m).$$

The previous discussion shows that by choosing the constants appropriately, under $\mathcal{E}_0, \mathcal{E}_1$, the policy $\pi_K$ satisfies $||\widehat{V}^{\pi_K} - V^\star||_\infty \le (1-\gamma)\varepsilon/12$, which implies that $||V^{\pi_K} - \widehat{V}^{\pi_K}||_\infty \le \varepsilon/3$. Thus, we have that

$$||V^\star - V^{\pi_K}||_\infty$$
$$\le ||V^\star - \widehat{V}^\star||_\infty + ||\widehat{V}^\star - \widehat{V}^{\pi_K}||_\infty + ||\widehat{V}^{\pi_K} - V^{\pi_K}||_\infty \le \varepsilon.$$
□

# C OMITTED DETAILS FROM SECTION 5

## C.1 Description and Analysis of the Mechanism from Lemma 5.3

In this subsection, we discuss the mechanism described in Lemma 5.3 and prove the lemma.

First, we provide a high-level overview of Mechanism 4 and explain some of the intricacies in its design. Let $x^*$ refer to the optimal MDP mechanism described in Corollary 3.4. We begin by ignoring the bidders in $S_2$ and first describe the allocation to bidders in $S_1$. We will return to how we allocate to bidders in $S_2$ at the end of our high-level discussion.

*Sketch of our proof of Lemma 5.3.* One can think of each of the bidders in $S_1$ as facing a reserve price set using their own modified virtual value and the highest modified virtual value from the bidders in $S_2$. Note that if $|S_1| = 1$ then this turns out to be very straightforward. In $x^*$, the sole bidder in $S_1$ is facing a threshold price from its own reserve and the competition from $S_2$. So, in Mechanism 4, we simply set this as its reserve price. This extracts the same revenue from all bidders in $S_1$ as $x^*$ does.

From now on, we assume that $|S_1| \ge 2$ and our goal now is to show that Mechanism 4 can extract at least $(1 - 1/|S_1|) \ge 1/2$ fraction of the revenue from $S_1$ that $x^*$ extracts. At this point, we note a technical issue that arises. If the modified virtual value for a bidder $i$ is constructed by adding a *positive* correction term, it may be that when advertiser $i$ wins, their (unmodified) virtual value is negative. Constructing a simple auction using negative virtual value reserves is not something that needed to be addressed in prior works on simple one-shot auctions, and turns out to be slightly tricky. To fix this, in the mechanism, we consider an auxiliary auction $x'$ which is identical to $x^*$ except (i) it does not allocate to bidders in $S_2$

and (ii) it sets an additional reserve to bidders in $S_1$ to ensure they only win when their (unmodified) virtual value is non-negative. This also had the additional benefit that the revenue that $x'$ extracts from $S_1$ is no less than the revenue that $x^*$ extracts from $S_1$. We remark that even though $x'$ extracts at least as much revenue from $S_1$ as $x^*$, it may not be that $x'$ is an optimal mechanism. The reason is that $x'$ allocates less often than $x^*$ and so the MDP transitions under $x'$ and $x^*$ are different.

At this point, we could turn $x'$ into a simple auction for bidders in $S_1$ by running a second price auction with the reserves used in $x'$. This would yield a simple auction which allocates to an advertiser in $S_1$ with exactly the same probability as $x'$ does and 2-approximate the contribution from $S_1$. The only issue is that $x'$ may allocate to a bidder in $S_1$ with probability *less* than $x^*$. So, we must somehow increase the probability that $x'$ allocates to a bidder. We do this as follows. Recall that we assume $|S_1| \geq 2$. From $S_1$, we remove the bidder that is contributing the least amount of revenue according to $x'$. Since we remove the bidder that is contributing the least revenue, we only lose at most $1/2$ of the revenue. Let $i^*$ denote the identity of the removed bidder. Now observe that, if we reintroduce this bidder with a reserve of 0 then we allocate to a bidder in $S_1$ with probability 1. Analogously, if we set an infinite reserve for $i^*$ then we allocate to a bidder in $S_1$ with probability no more than $x'$ did. However, we can tune this reserve so that the probability, over the value distribution, with which we allocate to a bidder in $S_1$ exactly matches that of $x^*$. Moreover, as we shall prove in Lemma C.2, running a second price auction with these new reserves extracts at least $1/2$ revenue that $x'$ does from $S_1 \setminus \{i^*\}$ which in turn is at least $1/4$ of the revenue that $x'$ extracts from $S_1$. We thus conclude that the second price auction with these new reserves 4-approximate the revenue from $S_1$ (according to $x^*$).

The above describes the first stage of the two-stage second-price auction. At this point, we have matched the probability of allocating to $S_1$ and have extracted good revenue from $S_1$. It remains to match the probability of allocating to $S_2$. For this, with a well-chosen probability, we run a second price auction with no reserves for the bidders in $S_2$. This probability is chosen to guarantee that Mechanism 4 allocates to some bidder in $S_2$ with the same probability as $x^*$.

We split the proof of Lemma 5.3 into parts. Lemma C.1 shows that allocation of Mechanism 4 matches that of $x^*$ and Lemma C.3 gives the revenue guarantee for Mechanism 4.

LEMMA C.1. *Let $S_1, S_2$ be disjoint sets such that $S_1 \cup S_2 = [n]$. Then Mechanism 4, given input $S_1, S_2$, outputs an ad in $S_1$ (resp. $S_2$) with exactly the same probability as the optimal MDP mechanism described in Corollary 3.4.*

PROOF. We focus on the case where $|S_1| \geq 2$ since if $|S_1| = 1$ then the only bidder in $S_1$ is facing the same (random) reserve in Mechanism 4 as it is in $x^*$.

Let $i^*$ be as defined in Line 13. Note that the allocation $x'$ defined in Line 12 allocates as long as some ad $i \in S_1$ exceeds their reserve of $r_i$ as defined in Line 11. By definition of $r_i$, this probability is at most $p_1$. Next, $p'_1$ is the probability that some bidder in $S_1 \setminus \{i^*\}$ exceeds their reserve $r_i$ which is at most the $p_1$. In Line 16, we set the reserve of $i^*$ so that the probability that advertiser $i^*$ fails to meet their reserve is $1 - \rho_1 = (1 - p_1)/(1 - p'_1)$. Therefore

---

**Mechanism 4** Two-stage second-price auction with $(S_1, S_2)$

1: **Input**: Value distribution $\mathcal{F}_i$ for each bidder $i$, current state ctr, disjoint sets $S_1, S_2$ such that $S_1 \cup S_2 = [n]$, optimal mechanism $x^*$ from Corollary 3.4.
2: **Output**: A two-stage second-price auction with reserves that allocates to $S_1$ (resp. $S_2$) with the same probability as the optimal mechanism from Corollary 3.4.
3: Let $R_j = \sum_{i \in S_j} \mathbb{E}_v \left[ x_i^*(v) \phi_i(v_i) \right]$ for $j \in \{1, 2\}$.
4: Let $p_j = \sum_{i \in S_j} \mathbb{E}_v \left[ x_i^*(v) \right]$ for $j \in \{1, 2\}$.
5: For each $i \in S_2$, sample $\widetilde{v}_i \sim F_i$.
6: Let $\widetilde{\phi} = \max_{i \in S_2} \widetilde{\phi}_i(\widetilde{v}_i, \text{ctr})$ and

$$\Delta_i = \gamma \left( \mathbb{E}_{\text{ctr}' \sim P_i(\cdot | \text{ctr})} \left[ V^*(\text{ctr}') \right] - \mathbb{E}_{\text{ctr}' \sim P_0(\cdot | \text{ctr})} \left[ V^*(\text{ctr}') \right] \right).$$

7: **if** $|S_1| = 1$ **then**
8:   Let $i$ be the only bidder in $S_1$ and set $r_i = \phi_i^{-1}(\max\{-\Delta_i, \widetilde{\phi} - \Delta_i\}/\text{ctr})$ as its reserve price.
9:   Allocate to bidder $i$ if they meet their reserve.
10: **else**
11:   Let $r_i = \phi_i^{-1}(\max\{-\Delta_i, \widetilde{\phi} - \Delta_i, 0\}/\text{ctr})$.
12:   Define the allocation $x_i'(v) = x_i^*(v) \cdot \mathbb{I}[i \in S_1 \wedge \phi_i(v_i) \geq 0]$.
13:   Let $i^* \in \text{argmin}_{i \in S_1} \{\mathbb{E}_v \left[ x_i'(v) \phi_i(v_i) \right]\}$.
14:   Let $p'_1 = \Pr[\exists i \in S_1 \setminus \{i^*\}, v_i \geq r_i]$.
15:   Let $\rho_1 = 1 - (1 - p_1)/(1 - p'_1)$.
16:   Redefine $r_{i^*} = F_{i^*}^{-1}(1 - \rho_1)$.
17:   Run a second price auction with reserves $r_i$ for all advertisers $i \in S_1$.
18: **end if**
19: If there is no winner so far then with probability $p_2/(1 - p_1)$ run a second price auction with no reserves for all advertisers $i \in S_2$.

---

the probability that some advertiser in $S_1$ exceeds their reserve is exactly $1 - (1 - \rho_1)(1 - p'_1) = p_1$. This latter probability is exactly that probability that an ad is shown by $x^*$ and by the second price auction in Line 17. Finally, note that by Line 19, an ad in $S_2$ is shown with probability $p_2$. □

Before we prove Lemma C.3, we require the following lemma. This lemma states that second price with reserves is a 2-approximation to the optimal mechanism with the same reserves. This can be viewed as an extension on previous results due to [23], [18], and [34] which prove this for the monopoly reserve.

LEMMA C.2. *Suppose we have $n$ independent bidders with virtual value functions $\phi_1, \ldots, \phi_n$. Let $r_1, \ldots, r_n$ be such that $r_i \geq \phi_i^{-1}(0)$ for all $i$. Let $z^*$ be the revenue-optimal auction that allocates to bidder $i$ only if their value is at least $r_i$. Then running a second price auction with reserve $r_i$ for bidder $i$ extracts at least $1/2$ the revenue as $z^*$.*

PROOF. Let $x$ be the allocation of the second price auction with reserve $r_i$ for bidder $i$. Let $\mathcal{E}_1 = \{v : x(v) = z^*(v)\}$ and $\mathcal{E}_2 = \{v :$

$x(v) \neq z^*(v)\}$. Then

$$\mathbb{E}_v\left[\sum_{i=1}^n x_i(v)\phi_i(v_i)\right] \geq \mathbb{E}_v\left[\sum_{i=1}^n x_i(v)\phi_i(v_i)\mathbb{I}[\mathcal{E}_1]\right]$$
$$= \mathbb{E}_v\left[\sum_{i=1}^n z_i^*(v)\phi_i(v_i)\mathbb{I}[\mathcal{E}_1]\right], \quad \text{(C.1)}$$

where the inequality is because if $v_i \geq r_i$ then $\phi_i(v_i) \geq 0$ and $x_i(v) = 1$ only if $v_i \geq r_i$.

Next, let $p_i(v)$ be the revenue generated by SPA with reserves $r_1, \ldots, r_n$ when the bidder values are $v = (v_1, \ldots, v_n)$. We claim that on the event $\mathcal{E}_2$, we have $\sum_{i \in [n]} p_i(v) \geq \sum_{i=1}^n z_i^*(v)v_i$. Indeed, if $\sum_{i=1}^n z_i^*(v) = 0$ then this is trivial. On the other hand, if $\sum_i z_i^*(v) \geq 1$ and $x \neq z^*$ then there must be at least two bidders that exceed their reserve (otherwise, we must have $x = z^*$ since there is only one eligible bidder). In SPA with reserves, the payment is at least the second-highest value which is at least the value of the advertiser that wins the auction according to $z^*$. Thus, $\sum_{i \in [n]} p_i(v) \geq \sum_{i=1}^n z_i^*(v)v_i$. Therefore,

$$\mathbb{E}_v\left[\sum_{i=1}^n p_i(v)\mathbb{I}[\mathcal{E}_2]\right] \geq \mathbb{E}_v\left[\sum_{i=1}^n z_i^*(v_i)v_i\mathbb{I}[\mathcal{E}_2]\right]$$
$$\geq \mathbb{E}_v\left[\sum_{i=1}^n z_i^*(v_i)\phi_i(v_i)\mathbb{I}[\mathcal{E}_2]\right], \quad \text{(C.2)}$$

where the last inequality is because value is always an upper bound on the virtual value. Note that the left-hand-side of both Eq. (C.1) and Eq. (C.2) give the value of SPA with reserves so combining the two equations completes the proof of the lemma. □

LEMMA C.3. *Fix a state s and disjoint sets $S_1, S_2$ such that $S_1 \cup S_2 = [n]$. The auction in Mechanism 4 obtains revenue at least $\frac{1}{4} \cdot \sum_{i \in S_1} \mathbb{E}_v\left[x_i^*(v)\phi_i(v_i)\right]$.*

Recall $\sum_{i \in S_1} \mathbb{E}_v\left[x_i^*(v)\phi_i(v_i)\right]$ is the revenue contributed by advertisers in $S_1$ in $x^*$.

PROOF. First, we claim that, with the allocation defined in Line 12, we have $\mathbb{E}_v\left[x_i'(v)\phi_i(v_i)\right] \geq \mathbb{E}_v\left[x_i^*(v)\phi_i(v_i)\right]$ for all $i \in S_1$. Indeed, for $i \in S_1$, we have

$$\mathbb{E}_v\left[x_i^*(v)\phi_i(v_i)\right] = \mathbb{E}_v\left[x_i^*(v)\phi_i(v_i)\mathbb{I}[\phi_i(v_i) < 0]\right]$$
$$+ \mathbb{E}_v\left[x_i^*(v)\phi_i(v_i)\mathbb{I}[\phi_i(v_i) \geq 0]\right]$$
$$\leq \mathbb{E}_v\left[x_i^*(v)\phi_i(v_i)\mathbb{I}[\phi_i(v_i) \geq 0]\right] = \mathbb{E}_v\left[x_i'(v)\phi_i(v_i)\right].$$

Hence the mechanism $x'$ obtains at least as much as revenue as $x^*$. We now show that the second price auction defined in Line 17 extracts at least a 1/4 of the revenue from $x'$ (and thus, from $x^*$).

First, suppose that there is only one bidder in $S_1$. Let $i$ be this single bidder. In this case, in $x^*$, bidder $i$ is just facing a threshold price set by its own reserve and the competition from $S_2$. In this case, Mechanism 4 sets this threshold as a reserve for bidder $i$.

On the other hand, suppose there are at least two bidders in $S_1$. Let $x''$ be the auction with the same allocation as $x'$ except $x_{i^*}''(v) = 0$ for all $v$. Then the auction $x''$ extracts at least $(1 - 1/|S_1|) \geq 1/2$ fraction of the revenue from $x'$. From Lemma C.2, a SPA auction with the same reserves $r_i$ for bidders in $S_1 \setminus \{i^*\}$ extracts at least 1/2 of the revenue as $x''$. Thus, SPA with reserves defined in Line 17

extracts at least 1/2 of the revenue as $x''$ so is at least 1/4 of the revenue as $x^*$. □

## C.2 Other Omitted Proofs from Section 5

Mechanism 5 describes the final mechanism. The following lemma shows that Mechanism 5 extracts at least 1/8 of the (present) revenue that $x^*$ extracts in each state while maintaining the same transitions as $x^*$.

LEMMA C.4. *Mechanism 5 shows a good (resp. bad) ad with exactly the same probability as $x^*$ and extracts at least 1/8-fraction of the revenue from $x^*$ at each state.*

PROOF. The lemma follows directly from Lemma 5.3 and the fact that $\max\{R_{\text{Good}}, R_{\text{BAD}}\}$ is already a 1/2-approximation to the revenue extracted from $x^*$ at each state. □

---

**Mechanism 5** Simple mechanism in each round of the MDP

1: **Input**: Value distribution $\mathcal{F}_i$ for each bidder $i$, current state ctr, qualities $q_1, \ldots, q_n$, optimal mechanism $x^*$ from Corollary 3.4.
2: Let $S_{\text{Good}} = \{i : q_i = 1\}$ and $S_{\text{BAD}} = \{i : q_i = -1\}$.
3: Let $R_{\text{Good}} = \sum_{i \in S_{\text{Good}}} \mathbb{E}_v\left[x_i^*(v)\phi_i(v_i)\right]$ and $R_{\text{BAD}} = \sum_{i \in S_{\text{BAD}}} \mathbb{E}_v\left[x_i^*(v)\phi_i(v_i)\right]$.
4: **if** $R_{\text{Good}} \geq R_{\text{BAD}}$ **then**
5:  Run Mechanism 4 (Appendix C.1) with $S_1 = S_{\text{Good}}$ and $S_2 = S_{\text{BAD}}$.
6: **else**
7:  Run Mechanism 4 (Appendix C.1) with $S_1 = S_{\text{BAD}}$ and $S_2 = S_{\text{Good}}$.
8: **end if**

---

We have proved that we can 8-approximate the revenue at each state and maintain the transition probability the same as the optimal mechanism. We now show how this implies an 8-approximation to the optimal MDP policy, thus proving the main theorem in this section.

PROOF OF THEOREM 5.1. Recall that the discounted reward for a policy $\pi$ is

$$\mathbb{E}_\pi\left[\sum_{t=0}^\infty \gamma^t R(\text{ctr}_t, a_t)\right] = \sum_{t=0}^\infty \gamma^t \mathbb{E}_\pi\left[R(\text{ctr}_t, a_t)\right]$$
$$= \sum_{t=0}^\infty \gamma^t \mathbb{E}_\pi\left[\mathbb{E}\left[R(\text{ctr}_t, a_t)|\text{ctr}_t\right]\right].$$

Let $\pi$ correspond to the policy by running Mechanism 5 in each state and let $\pi^*$ correspond to the optimal MDP policy from Corollary 3.4. Let $a_t$ (resp. $a_t^*$) denote the (random) auction chosen by $\pi$ (resp. $\pi^*$) at time $t$. Note that, conditioned on $\text{ctr}_t$, both $a_t$ and $a_t^*$ are actually deterministic. Then, by Lemma C.4, we have $\mathbb{E}\left[R(\text{ctr}_t, a_t)|\text{ctr}_t\right] \geq \frac{1}{8}\mathbb{E}\left[R(\text{ctr}_t, a_t^*)|\text{ctr}_t\right]$. Since the distribution of $\text{ctr}_t$ under policy $\pi$ or $\pi^*$ is the same, taking expectations on both sides completes the proof. □

Received 20 February 2007; revised 12 March 2009; accepted 5 June 2009

