# OpenReview forum: "User Response in Ad Auctions: An MDP Formulation of Long-term Revenue Optimization"
_ACM.org/TheWebConf/2024/Conference — TheWebConf24_

### Official Review · Reviewer_nqj1 · 2023-11-19

**Novelty:** 5
**Technical Quality:** 5

**Review:**

### Summary

The paper studies a dynamic online advertising auctions model where the quality of the ads shown in the past can affect the CTR of the user in the future.  In each stage, we have a number of identical slots for sale and each bidder wants at most one.  The novel component of the model is that there is a CTR that changes over time depending on which bidders win in each stage, and the values of all bidders are scaled down by this CTR in the same way.  So intuitively, the auctioneer (who cares about the discounted total revenue in the long run) wants to balance between extracting revenue in each stage and keeping the CTR high.

The main results are (1) a characterization through virtual values of the revenue optimal mechanism in the long run, (2) sample complexity results for learning approximately optimal mechanisms, and (3) a simple mechanism that gets a constant fraction of the optimal revenue (subject to additional assumptions).


### Strengths

The paper takes an attempt at an understudied phenomenon in online advertising.  The results are fairly comprehensive and inspiring.  The main technical novelty seems to be in the simple constant-approximation mechanism.


### Weaknesses

A number of results appear to follow from prior work in a relatively direct way.  A minor practical criticism is the proposed solution requires keeping track of each single user, which is computationally costly and poses potential privacy issues.

**Questions:**

Line 191: "second second-highest bid"

Line 296, "for simplicity, we assume the advertisers are myopic": I think this is a fair assumption, but I want to be sure what it means.  Does it mean "the results pretty much work for non-myopic bidders but that would complicate the presentation", or "it's unclear what would happen with non-myopic bidders"?  It would be nice to be clear in any case.

Footnote 6: the assumption does seem wlog in this paper, but I'm a bit confused -- does this mean all those models that explicitly model ad-specific CTR are redundant?

Thm 3.1: why do you need to handle the empty set differently?  I feel the theorem would look better if you simply said "choose W (of size between 0 and k, inclusively) that maximizes revenue in the current round + change to expected revenue in the future", and in particular you would be able to fit Eq (3.1) into a single line.

Line 500: "emprical MDP"

Sec 4: it's worth noting that you need a finite state space for all these results to work, which, technically speaking, contradicts what you say in the model section (line 324).  Also I suspect you could make things work with the state space being the [0, 1] interval, given some additional continuity assumptions on the transition operator.

Thm 5.1: while I believe this result is nontrivial and to some extent meaningful, I'm not sure it's "simpler" than the optimal auction.  (Essentially we are looking at two second-price auctions with personalized reserves, vs a single second-price auction with personalized reserves and personalized bid transformations.)

**Reviewer Confidence:**

3: The reviewer is confident but not certain that the evaluation is correct

**Scope:**

4: The work is relevant to the Web and to the track, and is of broad interest to the community

---

### Official Review · Reviewer_Q5st · 2023-11-22

**Novelty:** 6
**Technical Quality:** 6

**Review:**

# Summary

Motivated by the applications in the online advertising market, this paper introduces and studies a new auction model. In this model, there is a repeated multi-unit auction among myopic agents (advertisers) with linear utility. In each time period, there is a user-specific click-through-rate (CTR) that impacts agents' valuations and thus revenue. Here the CTR follows an MDP where the action is the winner of the auction (i.e., which subset of ads has been displayed). The goal is to design a IC, IR mechanism that maximizes the long-term discounted revenue.

The authors first characterized the optimal mechanism. It is a virtual surplus maximization mechanism, where the virtual value is the classic Myerson virtual value with an additive correction term that captures the future revenue. The authors then design a sample-efficient algorithm when the auction designer only has sample access to the CTR MDP and valuation distributions. Finally, the authors design a simple mechanism with a constant approximation guarantee.


# Strengths

This paper introduces a clear model with good motivation from real-world applications. The study is comprehensive and technically interesting. Overall, I enjoy reading this paper.

# Weaknesses

I don’t see significant weakness. I have some comments
discussed below.

# Comments

I wonder if the following argument leads to a 4-approximation simple mechanism (i.e., sequential posted pricing) in the single-slot environment. This argument does not require the assumption that the transition functions depend only on binary quality signals.

- For any state $ctr$ of the MDP, let $x^*_i(ctr)$ be the ex ante allocation probability of agent $i$ in the optimal mechanism. Consider implementing the Magician algorithm [Ala-14], or any single-item online contention resolution scheme [FSZ-16] as a sequential posted pricing (SPP) mechanism. Note this SPP ensures that each agent is allocated with probability $0.5 * x^*_i(ctr)$ and its revenue is a 2-approximation to the optimal mechanism in the current state $ctr$.

- Let $a$ be the agent that maximizes future revenue $E_{ctr'\sim P_i}[V^*(ctr')]$, and $b$ be the agent that has the second highest future revenue $E_{ctr'\sim P_i}[V^*(ctr')]$. Here $V^*$ is the value function for the optimal mechanism. First, we claim that if $E_{ctr'\sim P_a}[V^*(ctr')] <= E_{ctr'\sim P_0}[V^*(ctr')]$, then the constructed SPP above is already a 2-approximation to the optimal mechanism in the long run, since SPP essentially moves the probability of allocating to agents to the probability of no allocation, and no allocation leads to higher future revenue because of the if condition. A formal analysis may require an induction argument.

- If $E_{ctr'\sim P_a}[V^*(ctr')] >= E_{ctr'\sim P_0}[V^*(ctr')]$ and $E_{ctr'\sim P_b}[V^*(ctr')] <= E_{ctr'\sim P_0}[V^*(ctr')]$, then we modify the constructed SPP above by setting agent $a$ as the last one to approach and set the price such that her ex ante purchase probability is $x^*_a(ctr)$ instead of $0.5 x^*_a(ctr)$. Again, we claim that this modified SPP is a 2-approximation in the long run.

- Now, suppose $E_{ctr'\sim P_a}[V^*(ctr')] >= E_{ctr'\sim P_0}[V^*(ctr')]$ and $E_{ctr'\sim P_b}[V^*(ctr')] >= E_{ctr'\sim P_0}[V^*(ctr')]$. Consider two subcases.

- If the current state revenue contribution of agent $a$ is less than half of the total current state revenue in the optimal mechanism, then we modify the constructed SPP above by setting agent $a$ as the last one to approach and give the ad slot (if it is still available) to agent $a$ for free. We claim that this modified SPP is a 4-approximation in the long run.

- Finally, if the current state revenue contribution of agent $a$ is more than half of the total current state revenue in the optimal mechanism, then we know that the revenue contribution of agent $b$ is less than half of the total revenue. Here we modify the constructed SPP above by setting agent $a$ as the second last to approach with price such that her ex ante purchases probability is $x^*_a(ctr)$ instead of $0.5 x^*_a(ctr)$. Moreover, we set agent $b$ as the last agent to approach and give the ad slot (if it is still available) to agent $b$ for free. Again, we claim that this modified SPP is a 4-approximation in the long run.

Suppose discount factor $\gamma = 1 - O(1)$, do we know if the original Myerson auction (i.e., the one that maximizes current state virtual surplus) is a constant approximation to the optimal mechanism (characterized in Thm 3.1)?

In this paper, the authors assume that the ads' quality is public information. I think it would be interesting to explore if any results and techniques can be extended to the setting where the quality is also encoded as advertisers' private type. The optimal mechanism (by revelation principle, the one that asks agents to report both their value and quality) could be complicated, but there might exist simple mechanisms such as sequential posted pricing with constant good approximation.

*[Ala-14] Alaei, S. (2014). Bayesian combinatorial auctions: Expanding single buyer mechanisms to many buyers. SIAM Journal on Computing, 43(2), 930-972.*

*[FSZ-16] Feldman, M., Svensson, O., & Zenklusen, R. (2016). Online contention resolution schemes. In Proceedings of the twenty-seventh annual ACM-SIAM symposium on Discrete algorithms(pp. 1014-1033). Society for Industrial and Applied Mathematics.*

**Questions:**

I asked some questions/comments above but please don't feel obligated to respond.

**Reviewer Confidence:**

4: The reviewer is certain that the evaluation is correct and very familiar with the relevant literature

**Scope:**

4: The work is relevant to the Web and to the track, and is of broad interest to the community

---

### Official Review · Reviewer_AmaQ · 2023-11-23

**Novelty:** 5
**Technical Quality:** 4

**Review:**

The authors study an ad auction problem that tries to capture the user’s response to the quality of ads, and where the objective is the maximization of long-term revenue. In particular, the paper focuses on a repeated auction setting where n (myopic) advertisers are competing for k identical ad slots of a query from a single user in each round. It then suggests a formulation of a model that explores the interaction between the user and the advertisements as a Markov decision process (MDP). Under this formulation, the authors provide a characterization of the optimal mechanism as a Myerson’s auction, and they then propose simple and computationally efficient mechanisms that provide good approximation guarantees.

Strengths

The paper studies a very interesting and well-motivated problem. It tries to capture a complicated real-life scenario and it succeeds with the model that it proposes. On a technical level the paper is non-trivial, while in terms of clarity, as far as I checked, it is sound and correct.

Weaknesses

The introductory sections are not that well-written and, in my opinion, fail to describe the problem that the paper studies. The model is also presented in a confusing way, and it is not that easy to follow. Probably, some additional paragraphs giving some high level ideas should be included. Technically, the paper although non-trivial, it not that novel either and the analysis follows in general kind of standard approaches.


Overall, I would say that this is a paper that studies an interesting problem and provides a nice collection of results. It has however some problems in terms of presentation and structure, and would probably benefit from a revision (especially in the introductory sections).

**Questions:**

None.

**Reviewer Confidence:**

2: The reviewer is willing to defend the evaluation, but it is likely that the reviewer did not understand parts of the paper

**Scope:**

3: The work is somewhat relevant to the Web and to the track, and is of narrow interest to a sub-community

---

### Decision · Program_Chairs · 2024-01-22

**Decision:**

Accept

**Comment:**

The paper proposed a model for user behavior in ad auctions where their likelihood to click on an ad comes from an MDP. Within this model they characterize the revenue-optimal auction and they give a sample-efficient and computationally-efficient algorithm that yields the approximately optimal policy, and a simple constant-factor approximation algorithm.

 While the reviewers felt that the presentation could be improved, and some of the technical sections follow a roughly generally standard approach, the results were strong enough to warrant acceptance. Please include the reviewer suggestions in the camera ready version of the paper.